# CMTM6 suppresses cell-surface expression of death receptor FAS in mice but not in humans

Tereza Semberova[1], Michaela Pribikova[1], Veronika Cimermanova [2,3], Tijana Trivic [4], Rafik Haderbache[4], Darina Paprckova[4], Luca Christen [4], Helena Kissiova[1], Ondrej Stepanek [2] & Peter Draber [1,4 ✉]

## Abstract

The transmembrane protein CMTM6 promotes plasma membrane expression of the immune checkpoint protein PD-L1, a key suppressor of anti-tumor immunity. Targeting CMTM6 has been proposed as a strategy to enhance tumor cell killing by reducing PD-L1 surface expression. In accord, ablation of CMTM6 in mouse cancer models was shown to efficiently suppress tumor growth, but unexpectedly in a manner partially independent of PD-L1, suggesting that CMTM6 may regulate additional proteins involved in anti-tumor immunity. Using mass spectrometry, we discovered that mouse CMTM6 strongly associates with the cell death receptor FAS and negatively regulates its surface expression in mice. Deletion of CMTM6 increases FAS plasma membrane localization and sensitizes murine cells to FAS ligand-induced cytotoxicity. However, the interaction between CMTM6 and FAS is absent in human cells due to the difference in three amino acids at the boundary of the FAS extracellular and transmembrane domains. Altogether, our findings urge caution when translating promising data regarding the targeting of CMTM6 from mouse cancer models to potential human therapies.

Keywords CMTM6; PD-L1; FAS; CD95; Cell Death
Subject Categories Autophagy & Cell Death; Cancer; Immunology

## Introduction

Induction of programmed cell death is critical to maintain immune homeostasis and to remove damaged or infected cells. FAS (also known as CD95) is a widely expressed transmembrane receptor that can trigger apoptotic cell death upon binding of FAS ligand (FASL, also known as CD95L). In contrast, FASL is expressed mainly by activated T cells and NK cells (Nagata, 1997). FAS and FASL are important in controlling lymphocyte numbers. The deficiency in FAS or FASL proteins causes autoimmune lymphoproliferative syndrome (ALPS) in humans (Rieux-Laucat et al, 2018). In accord, ablation of either protein triggers severe lymphoproliferative disease in mice (Adachi et al, 1995; Karray et al, 2004; Takahashi et al, 1994; Watanabe-Fukunaga et al, 1992). Apart from regulating immune homeostasis, FAS-induced apoptosis is also employed by the immune system to kill virus-infected or transformed cells. Upon stimulation, the FAS intracellular death domain assembles a death-inducing signaling complex (DISC) containing adaptor FADD that recruits caspase-8, caspase-10, and their regulator cFLIP. Autoproteolytic cleavage of caspase-8/-10 releases their active forms into the cytoplasm, thus initiating the caspase cascade, and leading to apoptosis (Peter et al, 2015).

FAS stimulation can also trigger non-apoptotic signaling, especially activation of NF-κB and production of proinflammatory cytokines (Cullen et al, 2013; Davidovich et al, 2023; Siegmund et al, 2017). While immune cells can use FASL to kill tumor cells, some cancers are highly resistant to FAS-induced cell death and instead employ FAS-induced signaling to enhance their growth and invasiveness (Barnhart et al, 2004). Although the precise mechanism guiding the outcome of FAS stimulation is still under investigation, internalization of FAS appears crucial for the induction of cell death (Lee et al, 2006; Magri et al, 2024). Given the ability of FAS to trigger either cell death or activation and migration of cells, both FAS agonists and antagonists are being tested as therapeutic strategies to suppress tumor growth and modulate inflammation (Risso et al, 2022).

Immune surveillance of tumors involves T cells that can recognize and destroy cancer cells. However, their activity within the tumor microenvironment is often suppressed by the immune checkpoint protein programmed death-1 (PD-1), which binds to its ligand PD-L1, commonly expressed on cancer cells. Stimulation of PD-1 suppresses T cell activity, leading to their exhaustion (Morad et al, 2021). The surface expression of PD-L1 is regulated by association with small four-membrane domain-containing protein CMTM6 and, to a lesser degree, its homolog CMTM4 (Burr et al, 2017; Dai et al, 2021; Mezzadra et al, 2017). Indeed, targeting CMTM6 in several mouse cancer models led to enhanced anti-tumor cytotoxicity, which was only partially caused by decreased PD-L1 expression, since blocking PD-L1 together with CMTM6 depletion had a substantially stronger effect than each treatment

[1]Laboratory of Immunity & Cell Communication, Division BIOCEV, First Faculty of Medicine, Charles University, Vestec, Czech Republic. [2]Laboratory of Adaptive Immunity, Institute of Molecular Genetics of the Czech Academy of Sciences, Prague, Czech Republic. [3]Faculty of Science, Charles University, Prague, Czech Republic. [4]Department of Immunobiology, University of Lausanne, Epalinges, Switzerland. ✉E-mail: peter.draber@unil.ch

alone (Long et al, 2023). This indicated that in addition to enhancing PD-L1 surface expression, CMTM6 protects mouse cancer cells by regulating the activity or expression of other protein(s).

Here, we aimed to identify the interacting partners of mouse CMTM6 compared to other CMTM family members to search for other immunologically relevant proteins they might regulate. We showed that endogenous CMTM6 strongly binds to mouse FAS and suppresses its plasma membrane localization, therefore reducing sensitivity to FASL-driven apoptosis. However, human FAS does not bind CMTM6. This is caused by a difference in three amino acids in the transmembrane and extracellular region between human and mouse FAS. Altogether, our data demonstrate different regulation of FAS membrane expression between mouse and human cells and suggest that targeting human CMTM6 might not have the same benefit as targeting the protein in mouse tumor models.

## Results

### Mouse FAS is strongly associated with CMTM6

CMTM4 and CMTM6 proteins were previously reported as important regulators of immune responses and anti-tumor immunity because they directly bind and regulate plasma membrane localization of several proteins, such as PD-L1 (Burr et al, 2017; Dai et al, 2021; Mezzadra et al, 2017), IL-17 receptor C (IL-17RC) (Knizkova et al, 2022; Ni et al, 2024), or the epidermal growth factor receptor (EGFR) (Xu et al, 2025). In here, we

employed mass spectrometry to elucidate whether CMTM4 and CMTM6 bind other immunologically relevant proteins that might play a role in regulating tumor immunosurveillance.

We first prepared retroviral vector coding for CMTM4 fused at the C-terminus to the 2xStrep 3xFlag (SF) tag. Expression of CMTM4-SF in $Cmtm4^{KO}$ mouse stromal ST2 cells fully rescued the expression of surface IL-17RC, indicating that the addition of the tag does not interfere with CMTM4 function (Fig. EV1A). Next, we prepared six different ST2 cell lines expressing CMTM3 to CMTM8 fused to the SF-tag. We excluded CMTM1 and CMTM2 from our study because these proteins are testis-specific according to the Human Protein Atlas (Uhlen et al, 2015) and unstable upon overexpression in ST2 cell line (Knizkova et al, 2022). We purified individual CMTM family members using tandem affinity purification and analyzed associated proteins by mass spectrometry (Dataset EV1). Protein quantification was based on Top3 intensity, which sums the three most intense peptides derived from each protein. We identified PD-L1 among the top 20 interactors of CMTM6 (Fig. 1A). Similarly, IL-17RC was a very strong interactor of CMTM4 (Fig. EV1B), which validated our experimental approach. Some members of the CMTM family interacted together, such as CMTM4 and CMTM6, suggesting they might create membrane nanodomains, as was observed for proteins of the tetraspanin family (van Deventer et al, 2021) (Fig. EV1C). Importantly, we observed that cell-death receptor FAS strongly interacted with CMTM6 and was very weakly associated with CMTM4 but not with other CMTM family members (Fig. 1A,B). Finally, we confirmed the strong interaction between CMTM6-SF and murine FAS via immunoblotting (Fig. 1C).

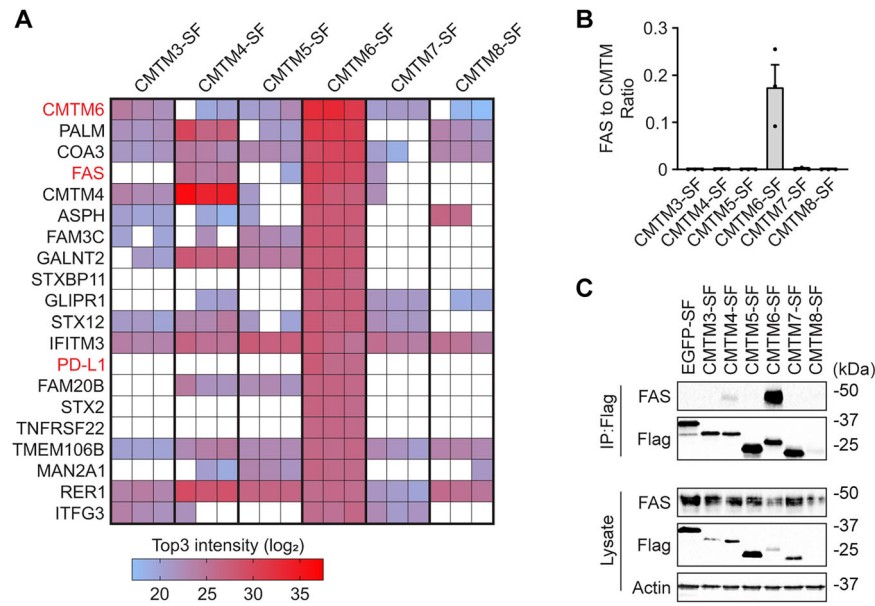

**Figure 1. CMTM6 associates with FAS in mouse cells.**

(A) Mass spectrometry analysis of the indicated Strep-Flag (SF)-tagged CMTM family members isolated from mouse ST2 cells via tandem affinity purification. The heatmap shows the most abundant CMTM6 interactors that were not detected in control EGFP-SF samples, based on Top3 intensities from 3 independent experiments. (B) Quantification of the Top3 intensity ratio of FAS relative to individual SF-tagged CMTM family members. Data are presented as mean + SEM from 3 independent experiments. (C) Samples isolated via Flag immunoprecipitation from ST2 cells expressing the indicated SF-tagged CMTM family members or control EGFP-SF were analyzed by immunoblotting. Data are representative of 3 independent experiments. Source data are available online for this figure.

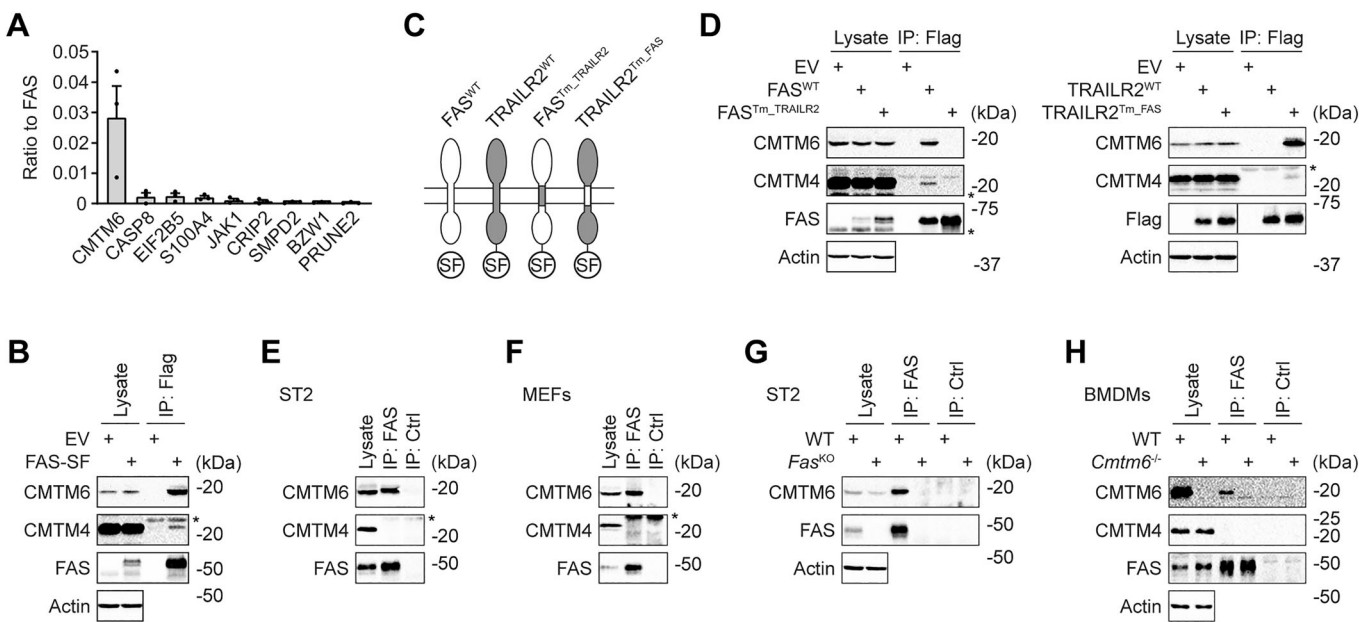

**Figure 2.  Mouse FAS binds strongly to CMTM6 via the transmembrane domain.**

(A) Mass spectrometry analysis of the SF-tagged mouse FAS isolated from mouse ST2 cells via tandem affinity purification. The most abundant FAS interactors not detected in control EGFP-SF samples are shown, based on Top3 intensities from 3 independent experiments. Data are presented as mean + SEM. (B) ST2 cells expressing SF-tagged mouse FAS or empty vector (EV) were lysed, subjected to Flag immunoprecipitation, and analyzed by immunoblotting. (C) Schematic representation of mouse SF-tagged FAS and TRAILR2 chimeric proteins expressed in ST2 cells. (D) SF-tagged FAS, TRAILR2, and the indicated chimeric proteins were purified via Flag immunoprecipitation and analyzed by immunoblotting. (E–H) Lysates from the indicated cell lines were subjected to immunoprecipitation with anti-FAS or isotype control antibodies and analyzed by immunoblotting. Data are representative of two (E–H), three (A, B), and five (D) independent experiments. *, nonspecific band. Source data are available online for this figure.

To validate our data, we expressed murine FAS fused to SF-tag at the C-terminus in ST2 cells and analyzed the FAS-interactome via mass spectrometry upon tandem affinity purification. We observed that in unstimulated cells, CMTM6 was the major interactor of FAS (Dataset EV2 and Fig. 2A). We confirmed their interaction using immunoblotting and we also detected very weak binding of CMTM4 (Fig. 2B). CMTM6 and CMTM4 have four transmembrane domains with short extracellular or cytoplasmic sequences, which indicated that they might bind to transmembrane or juxtamembrane domain of murine FAS. To test this hypothesis, we expressed a chimeric protein in which the FAS transmembrane and juxtamembrane parts were exchanged for the corresponding sequence from cell death receptor TRAILR2 that does not interact with CMTM family members (Figs. 2C and EV1D). This led to the complete loss of CMTM6 and CMTM4 binding (Fig. 2D). In contrast, TRAILR2 harboring the transmembrane and juxtamembrane domains of murine FAS strongly interacted with CMTM6, and we detected weak binding of CMTM4 (Fig. 2C,D). Altogether, these data showed that FAS interacted with both CMTM4 and CMTM6 via the transmembrane region or potentially via several amino acids flanking this domain.

The immunoprecipitation of endogenous FAS from lysates of mouse ST2 cells or mouse embryonic fibroblasts (MEFs) showed a strong interaction with CMTM6. We did not detect CMTM4, indicating that CMTM6 is the major interacting partner of endogenous protein (Fig. 2E,F). Furthermore, we co-immunoprecipitated CMTM6 using anti-FAS antibody from lysates of ST2 wild-type (WT) cells but not from FAS-deficient cells

(Fig. 2G), which confirmed the specificity of the interaction. Finally, anti-FAS antibody co-immunoprecipitated CMTM6 from lysates of bone marrow-derived macrophages (BMDMs) isolated from WT mice, but not from *Cmtm6*$^{-/-}$ animals (Fig. 2H), while CMTM4 was not detected. Combined, these data established a strong and specific interaction of endogenous FAS with CMTM6 in mouse cells.

## CMTM6 suppresses FAS membrane localization

CMTM6 was previously shown to promote PD-L1 plasma membrane localization. FAS is an important regulator of immune homeostasis, and FAS-deficient mice develop lymphoproliferation (Peter et al, 2015). To evaluate whether CMTM6 is important in regulating immune system homeostasis, we analyzed the T cell compartment in *Cmtm6*$^{-/-}$ mice (gating strategy is shown in Fig. EV2). However, we did not detect any significant changes. There were no differences in spleen weight or cellularity between WT and *Cmtm6*$^{-/-}$ mice (Figs. 3A and EV3A). The proportion of splenic CD3$^+$ T cells and the ratio between cytotoxic CD8$^+$ T cells, helper CD4$^+$ T cells, and regulatory T cells (Tregs) was unchanged (Fig. 3B). We did not detect an increased proportion of splenic memory CD4$^+$ T cells in *Cmtm6*$^{-/-}$ mice (Fig. 3C). Similarly, we observed no changes in the proportion of splenic memory CD8$^+$ T cells and antigen inexperienced memory-like CD8$^+$ T cells (AIMT) that did not encounter antigen (Moudra et al, 2021) (Fig. 3D). We observed the same results in lymph nodes (Fig. EV3B–D), indicating that CMTM6 is not required for the regulation of general T-cell homeostasis.

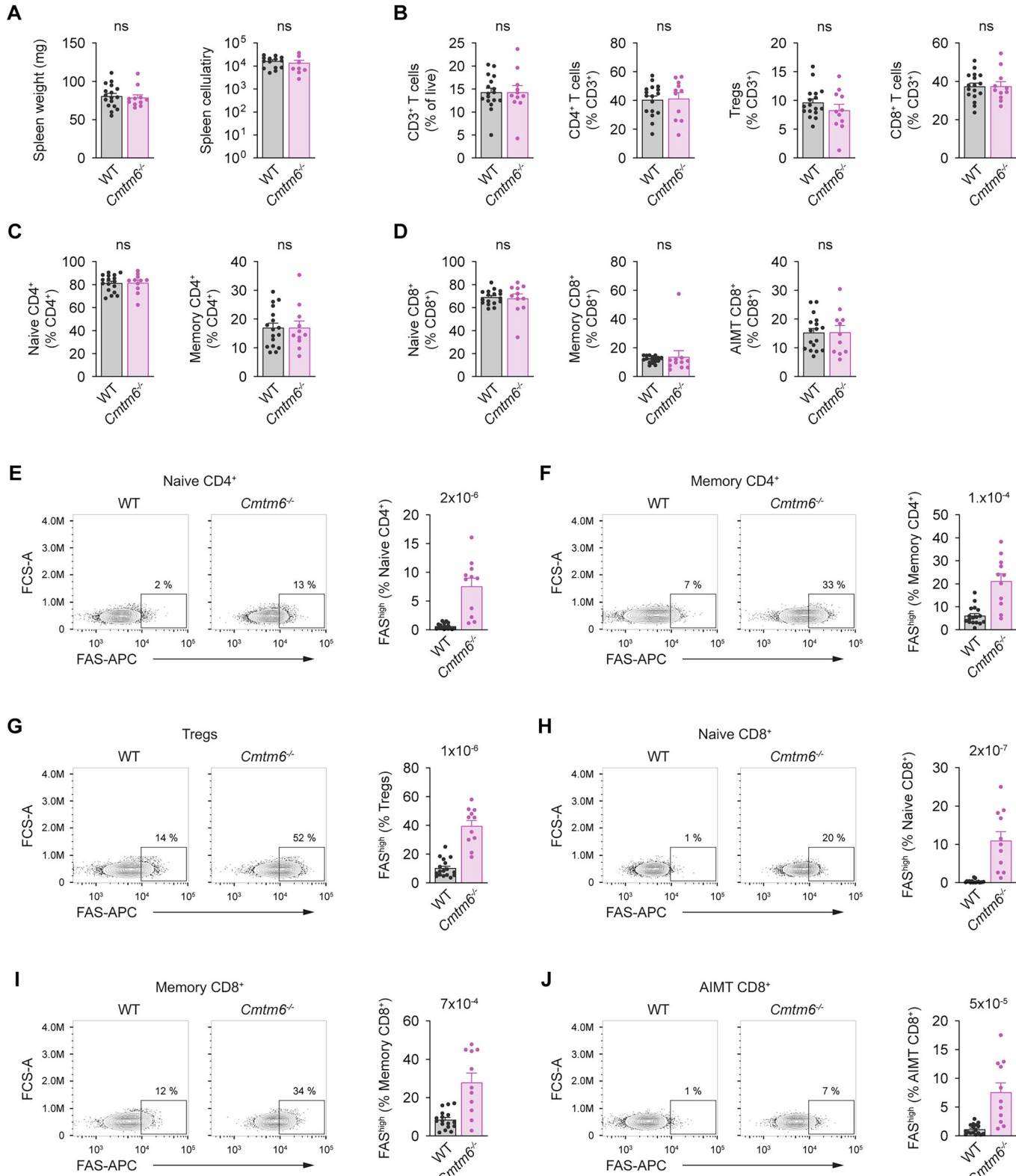

Surprisingly, we noted that $Cmtm6^{-/-}$ T cells have markedly elevated expression of surface FAS compared to WT cells. We observed highly increased surface expression of FAS in both naïve and memory CD4+ T cells (Fig. 3E,F), Tregs (Fig. 3G), and in naïve,

memory, and AIMT CD8+ T cells (Fig. 3H–J). Similarly, increased FAS surface expression was observed in different T cell subsets isolated from lymph nodes of $Cmtm6^{-/-}$ mice (Fig. EV3E–J). These results show that CMTM6 binding to FAS suppresses its membrane

**Figure 3.   CMTM6 suppresses FAS surface levels in mouse splenic T cells.**

(A) Weight and cellularity of spleens isolated from 8 to 12-week-old WT and *Cmtm6*$^{-/-}$ mice. (B) Flow cytometry analysis of splenic T cells from 8 to 12-week-old WT and *Cmtm6*$^{-/-}$ mice. T cells (gated as CD3$^+$) were subdivided into conventional CD4$^+$ (CD4$^+$, FOXP3$^-$), Tregs (CD4$^+$, FOXP3$^+$), and conventional CD8$^+$ (CD8$^+$). (C) Conventional CD4$^+$ T cells were further gated as naïve (CD44$^-$, CD62L$^+$) and memory (CD44$^+$, CD62L$^-$). (D) Conventional CD8$^+$ T cells were gated as naïve (CD44$^-$), memory (CD44$^+$, CD49d$^+$), and antigen-inexperienced memory T (AIMT) (CD44$^+$, CD49d$^-$) cells. (E–J) Surface FAS levels in the indicated subsets of splenic T cells. Data are presented as mean + SEM, n = 17 WT and 11 *Cmtm6*$^{-/-}$ mice. Two-tailed Mann-Whitney test. ns not significant. Source data are available online for this figure.

expression without affecting immune homeostasis. This was intriguing, as CMTM6 has an opposite effect on the immunosuppressive protein PD-L1 and promotes its membrane localization (Burr et al, 2017; Mezzadra et al, 2017).

## CMTM6 regulates FAS expression in the course of an infection

To elucidate whether CMTM6 suppresses FAS expression during an infection, WT and *Cmtm6*$^{-/-}$ mice were infected intravenously with *Listeria monocytogenes* expressing ovalbumin (OVA) antigen (Fig. 4A). After 10 days, we isolated the spleens and analyzed T cells via flow cytometry (Fig. 4B and gating strategy in Fig. EV4A). We noted a markedly decreased proportion of effector CD4$^+$ T cells (Fig. 4C), while the emergence of effector and OVA-specific CD8$^+$ T cells was not significantly impacted (Fig. 4D). Importantly, we noticed that, similarly to naïve cells, CD4$^+$ and CD8$^+$ effector and OVA-specific T cells isolated from *Cmtm6*$^{-/-}$ mice had substantially increased expression of surface FAS compared to WT mice (Fig. 4E–I). In contrast, the surface levels of FASL were not different between WT and *Cmtm6*$^{-/-}$ mice (Fig. EV4B–F). These data further demonstrate that CMTM6 is a potent negative regulator of FAS expression in murine T cells.

## CMTM6 suppresses FAS-induced cell death in mouse cells

To validate that CMTM6 suppressed FAS expression on the cell surface, we isolated MEFs or BMDMs from WT or *Cmtm6*$^{-/-}$ littermates. Indeed, the surface FAS was substantially increased in *Cmtm6*$^{-/-}$ cells (Figs. 5A and EV5A). Stimulation of *Cmtm6*$^{-/-}$ MEFs with hexameric FAS ligand (Hex-FASL) led to markedly enhanced induction of apoptosis compared to WT controls (Fig. 5B). In accord, we detected increased cleavage of caspase-8 and caspase-3 in *Cmtm6*$^{-/-}$ cells upon Hex-FASL stimulation compared to WT cells (Fig. 5C).

As expected, reconstitution of *Cmtm6*$^{-/-}$ MEFs with CMTM6 led to decreased FAS expression compared to cells reconstituted with empty vector (EV) only (Fig. 5D). In accord, enhanced induction of cell death and activation of caspase-8 and caspase-3 upon FASL stimulation in *Cmtm6*$^{-/-}$ MEFs were rescued upon reconstitution of cells with CMTM6, but not EV (Fig. 5E,F). Altogether, our data demonstrate that CMTM6 suppresses FAS expression on the surface of mouse cells, which correlates with reduced FASL-induced activation of apoptosis.

To elucidate whether CMTM6 promotes FAS internalization, cells were labeled on ice with anti-FAS antibody and subsequently incubated at 37 °C for several hours. The remaining surface FAS was detected using a fluorescently labeled secondary antibody. *Cmtm6*$^{-/-}$ MEFs had a substantially decreased rate of FAS

internalization compared to WT cells (Fig. 5G). We confirmed these data by showing that *Cmtm6*$^{-/-}$ MEFs reconstituted with CMTM6 had enhanced FAS internalization compared to cells transduced with EV (Fig. EV5B). Finally, we expressed CMTM6-mCherry and FAS-EGFP in *Cmtm6*$^{-/-}$ cells and showed the colocalization of both proteins in recycling endosomes, detected by staining with transferrin receptor (Fig. 5H). Altogether, our data indicate that CMTM6 enhances FAS internalization and suppresses its surface expression.

## CMTM6 does not interact with human FAS

Since our data indicated that CMTM6 has an important role in regulating FAS in mouse cells, we aimed to validate our findings in human cells. To our surprise, we detected no interaction between endogenous human FAS and CMTM6 in HeLa cells (Fig. 6A) or in HEK293T cells (Fig. 6B). To exclude that technical issues had caused the lack of interaction between the two proteins in human cells, we expressed human FAS or PD-L1 fused to SF-tag in HeLa cells. Immunoprecipitation of cell lysates using anti-Flag antibody revealed a strong interaction of CMTM6 with overexpressed human PD-L1 but not with human FAS (Fig. 6C). Altogether, these data indicated that FAS and CMTM6 do not interact in human cells. Mouse and human CMTM6 are very similar, sharing 83% of identical amino acids (Fig. EV5C). This suggests that the differences between human and mouse FAS regarding CMTM6 binding are likely due to the differences in the amino acid composition of FAS transmembrane and juxtamembrane domains.

We performed a series of experiments in which we swapped parts of human FAS for corresponding sequences in mouse FAS to identify what amino acid changes are causing the loss of CMTM6 binding to human FAS (Fig. 6D). First, we noted that human FAS$^{Mut1}$, in which we exchanged only the transmembrane domain with its mouse counterpart, was unable to bind CMTM6. In contrast, human FAS$^{Mut2,}$ in which we exchanged the transmembrane domain and the surrounding regions with mouse FAS, interacted strongly with CMTM6 in human HeLa cells (Fig. 6E). Subsequently, we demonstrated that human FAS$^{Mut4}$, in which we swapped only the transmembrane and proximal extracellular part with the mouse counterpart, enabled CMTM6 binding (Fig. 6F).

Next, we started changing amino acids in the mouse FAS transmembrane and proximal extracellular part to those found in human FAS. The FAS$^{Mut7-8}$, which harbored mutations of four amino acids at the boundary of the extracellular and transmembrane parts, lost the ability to bind CMTM6 (Fig. 6G). Finally, detailed mapping in FAS$^{Mut14-16}$ showed that three amino acids in mouse FAS that differ from those in humans are required for the interaction (Fig. 6H). Importantly, human FAS$^{Mut17}$ with four amino acid changes to mouse homologs, i.e., S172N, N173R, G175W, and W176L, fully regained the ability to bind CMTM6

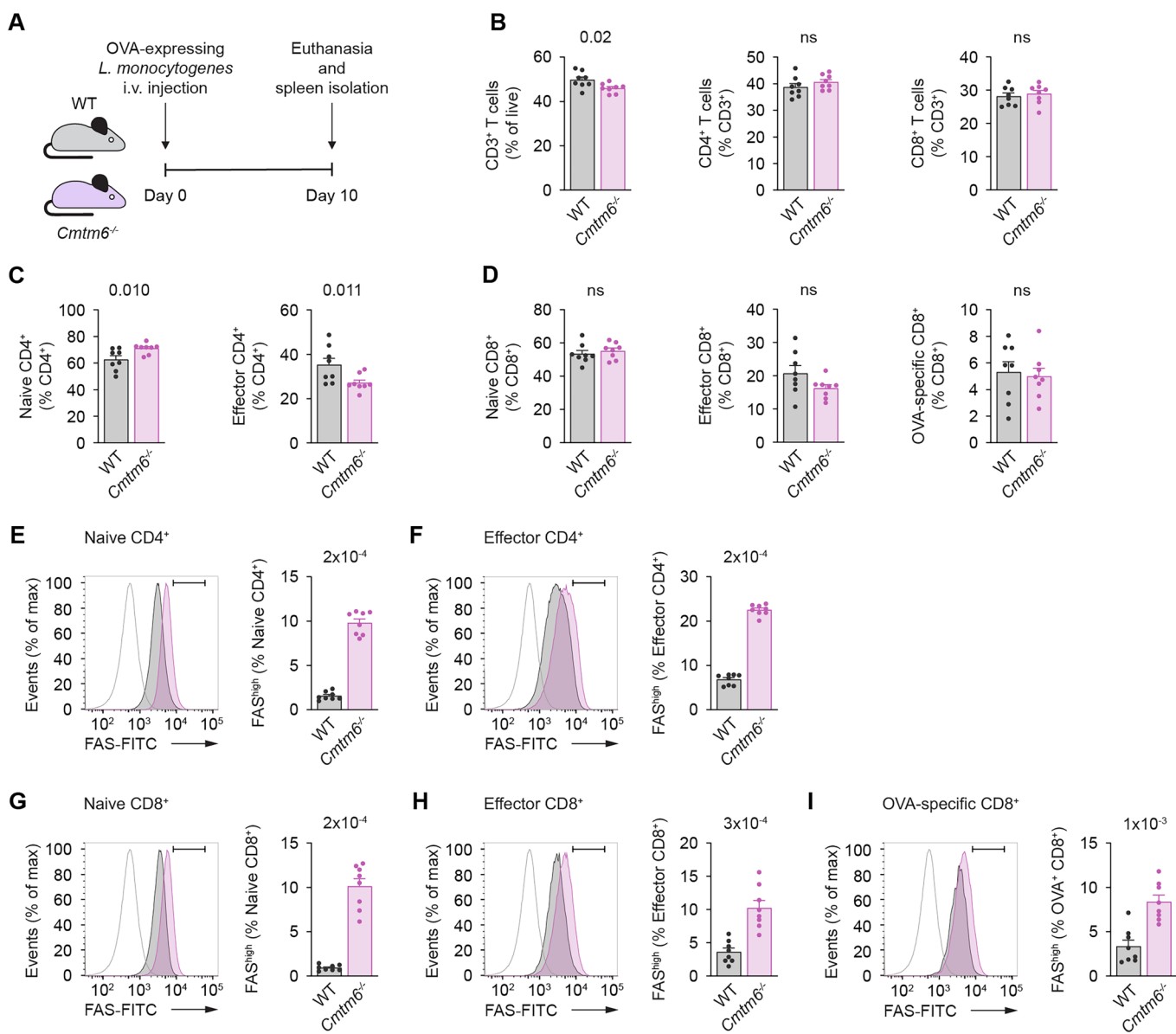

**Figure 4. CMTM6 regulates FAS expression in the course of *L. monocytogenes* infection.**

(A) Experimental design of *L. monocytogenes* infection. (B) Flow cytometry analysis of splenic T cells from 8-12-week-old WT and *Cmtm6$^{-/-}$* mice. T cells (gated as CD3$^+$) were subdivided into conventional CD4$^+$ and CD8$^+$. (C) CD4$^+$ T cells were further gated as naïve (CD44$^-$, CD62L$^+$) and effector (CD44$^+$, CD62L$^-$). (D) CD8$^+$ T cells were gated as naïve (CD44$^-$, CD62L$^+$), effector (CD44$^+$, CD62L$^-$), and OVA-specific (binding Kb-OVA tetramers). (E-I) Surface FAS levels in the indicated subsets of splenic T cells. Data are presented as mean + SEM from two independent experiments. $n = 8$ mice per group. Two-tailed Mann-Whitney statistical test. ns, not significant. Source data are available online for this figure.

(Fig. 6H). These results show that the extracellular and transmembrane domain boundary of mouse FAS is required for interaction with CMTM6. Since this sequence is different in human FAS, the interaction is lost.

The absence of interaction between human FAS and CMTM6 prompted us to elucidate whether human CMTM6 can regulate a different set of immunologically relevant proteins in human cells. To study the interactome of human CMTM6, we retrovirally expressed SF-tagged human CMTM6 in HeLa cells, isolated CMTM6-SF via tandem affinity purification, and analyzed the associated proteins via mass spectrometry (Dataset EV3). We

observed that the strongest interactor of human CMTM6 was the human immunoregulatory protein CD58 (Fig. EV5D). We did not detect PD-L1 in our samples, as its expression is induced only upon IFNγ stimulation in HeLa cells (Fig. EV5E). CD58 is an important activator of anti-tumor immunity, which is missing in rodents and was previously shown to interact with human CMTM6 (Ho et al, 2023; Miao et al, 2023). The association of CMTM6 with human FAS was not detected. In accord, CMTM6-deficient HeLa cells had decreased surface expression of CD58 and also decreased expression of PD-L1 in IFNγ-stimulated cells, while the expression of FAS was not changed (Fig. 7A). We observed similar kinetics of Hex-

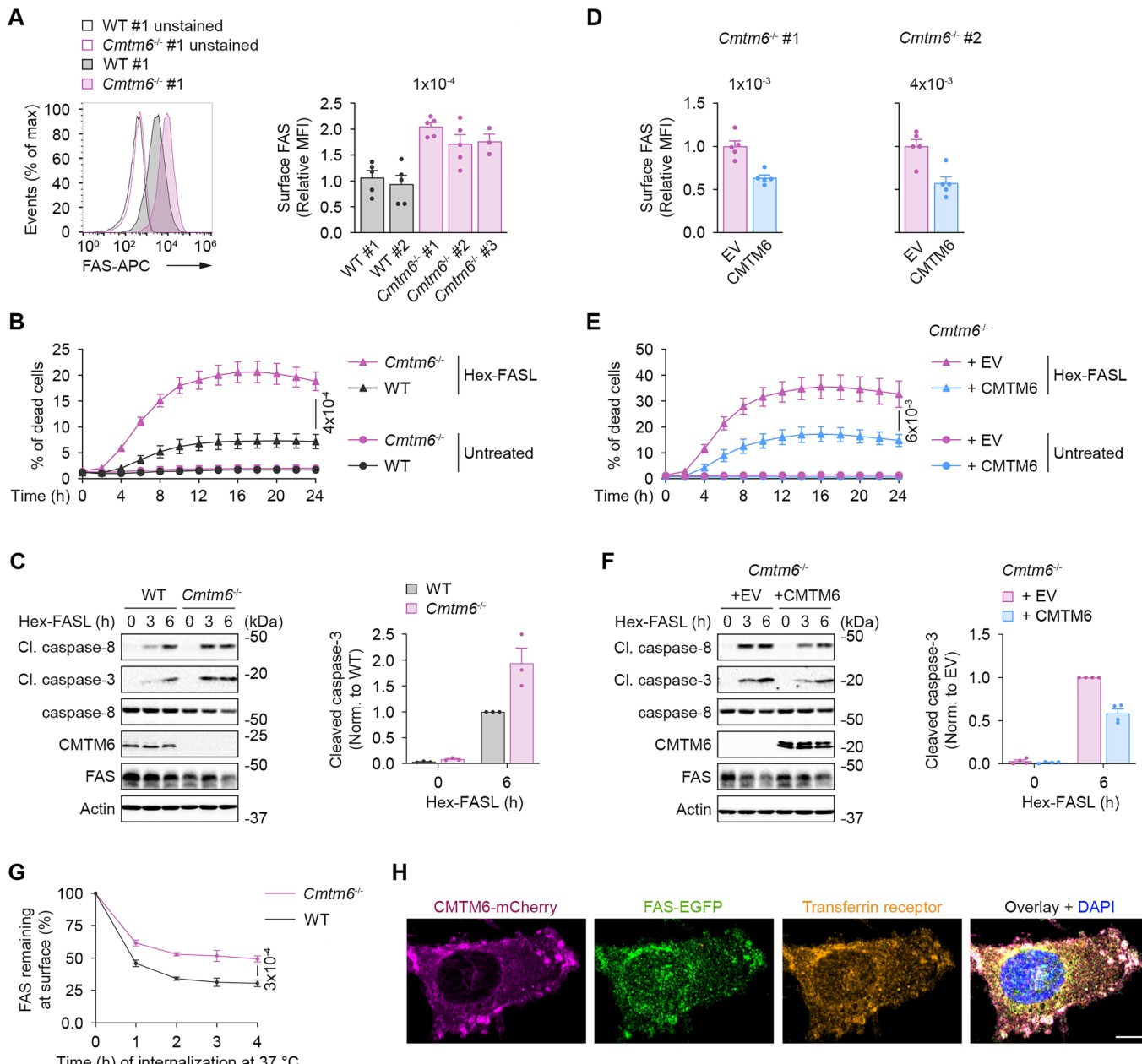

**Figure 5. CMTM6 suppresses FAS-induced cell death in mouse cells.**

(A) Flow cytometry analysis of surface FAS expression in MEFs isolated from WT and $Cmtm6^{-/-}$ sibling embryos. (B) Cell death induction in WT or $Cmtm6^{-/-}$ MEFs (clones #1 and #2) stimulated or not with Hex-FASL (500 ng/ml). Cell death was monitored every 2 h using Incucyte. (C) Immunoblot analysis of lysates from WT or $Cmtm6^{-/-}$ MEFs stimulated with Hex-FASL (500 ng/ml) for the indicated time points. (D) Flow cytometry analysis of surface FAS expression in $Cmtm6^{-/-}$ MEFs transduced with either an empty vector (EV) or a vector expressing untagged murine CMTM6. (E) Cell death induction in $Cmtm6^{-/-}$ MEFs (clones #1 and #2) reconstituted or not with CMTM6 that were stimulated or not with Hex-FASL (500 ng/ml). Cell death was monitored every 2 h using Incucyte. (F) Immunoblot analysis of lysates from $Cmtm6^{-/-}$ MEFs reconstituted or not with CMTM6 that were stimulated with Hex-FASL (500 ng/ml) for the indicated time points. (G) FAS internalization assay in WT or $Cmtm6^{-/-}$ MEFs (clones #1 and #2). Cells were labeled on ice with unconjugated FAS antibody and subsequently incubated for the indicated time at 37 °C. The remaining surface FAS was detected using a fluorescently labeled secondary antibody and analyzed by flow cytometry. (H) Confocal microscopy of sub-cellular localizations in MEFs transduced with CMTM6-mCherry and FAS-EGFP proteins and stained for transferrin receptor. Scale bar, 5 μm. Data are presented as mean + SEM (A, C, D, F) or mean ± SEM (B, E, G). Data are representative of three (C, G, H), four (F), five (A, D), and six (B, E) independent experiments. One-way ANOVA (A) and an unpaired two-tailed t-test (B, D, E, G). MFI median fluorescent intensity. Source data are available online for this figure.

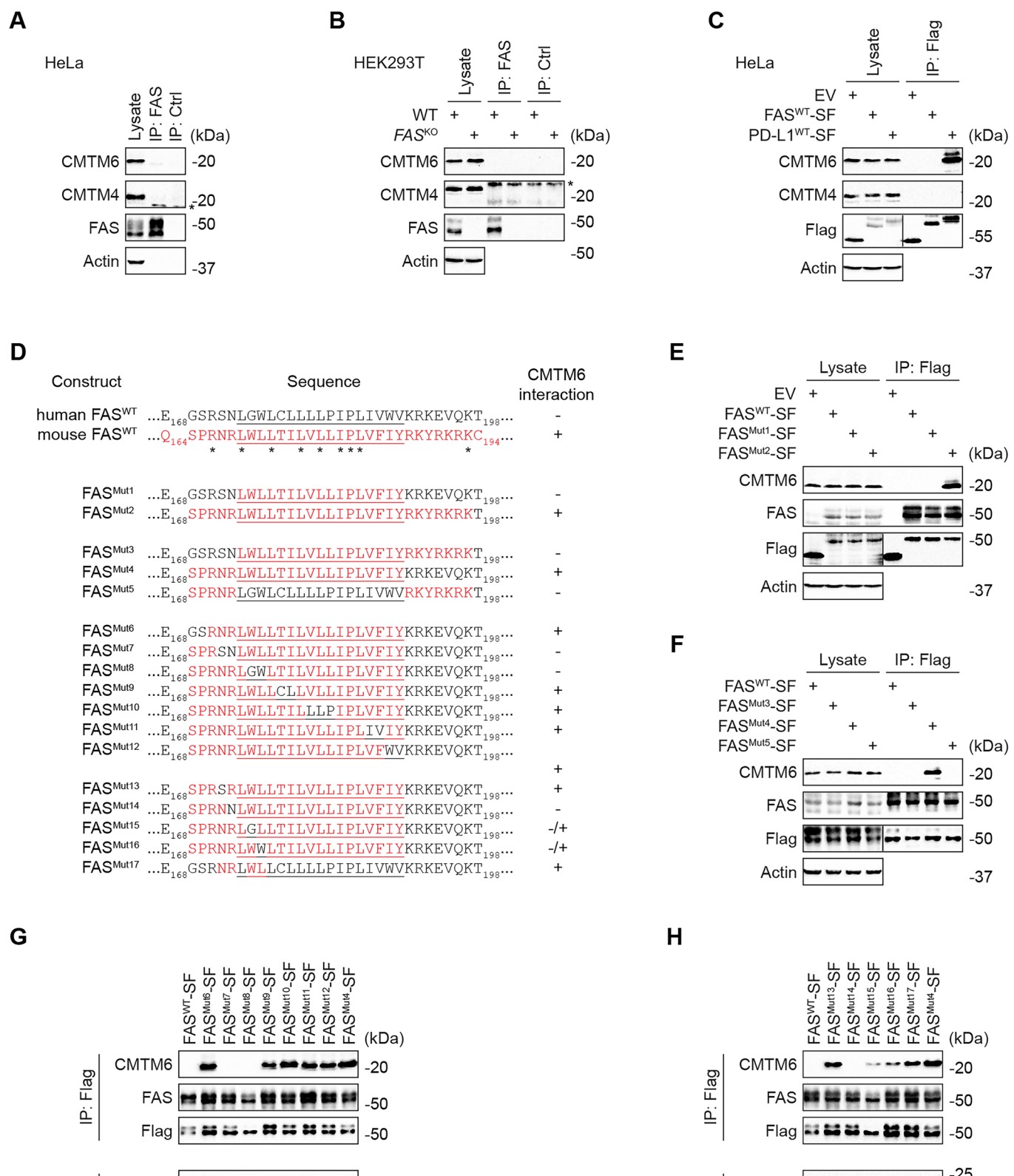

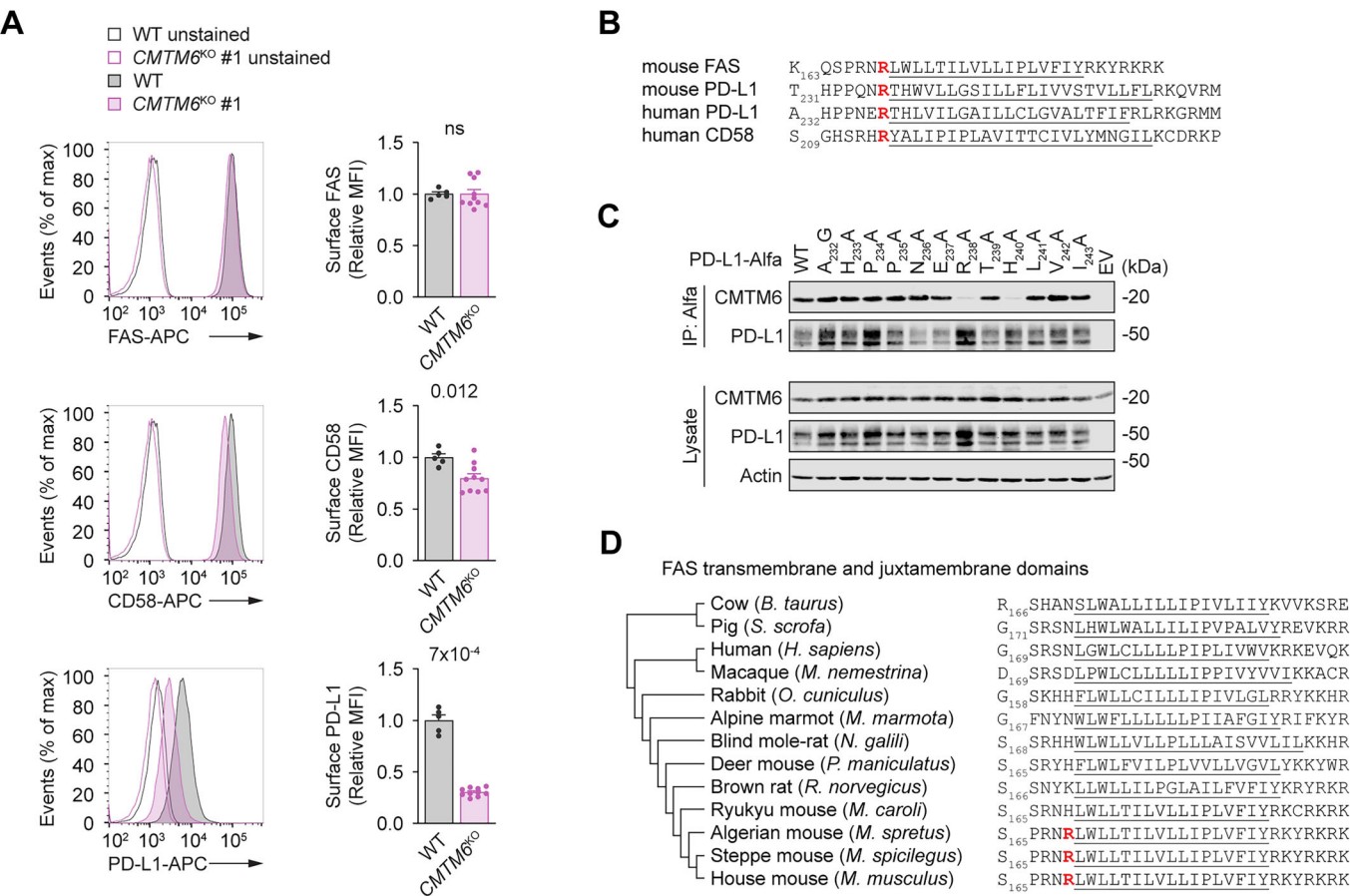

◀ **Figure 6.  Human FAS does not interact with CMTM6.**

(A, B) Lysates from the indicated cell lines were subjected to immunoprecipitation using anti-FAS or isotype control antibodies and analyzed by immunoblotting. (C) HeLa cells expressing SF-tagged human FAS, human PD-L1, or empty vector (EV) were lysed, subjected to Flag immunoprecipitation, and analyzed by immunoblotting. (D) Sequences of swap mutants between human and mouse FAS. The transmembrane domain is underlined. (E–H) Lysates from HeLa cells expressing the indicated SF-tagged FAS mutants were subjected to Flag immunoprecipitation and analyzed by immunoblotting. Data are representative of two (A, B, E–H) and three (C) independent experiments. *, nonspecific band. Source data are available online for this figure.

**Figure 7.  Human CMTM6 regulates surface expression of PD-L1 and CD58, but not FAS.**

(A) Flow cytometry analysis of surface FAS, CD58 in untreated and PD-L1 in IFNγ-stimulated (100 ng/ml) WT and *CMTM6*$^{KO}$ HeLa cells (two distinct clones). Data are presented as mean + SEM. (B) Sequence comparison of transmembrane and juxtamembrane domains of the selected proteins. (C) Lysates from HeLa cells expressing the indicated Alfa-tagged PD-L1 mutants were subjected to anti-Alfa immunoprecipitation and analyzed by immunoblotting. (D) Comparison of FAS transmembrane (underlined) and juxtamembrane domains in indicated species. Data are representative of three (A, C) independent experiments. Two-tailed Mann-Whitney test (A). ns not significant, MFI median fluorescent intensity. Source data are available online for this figure.

FASL-induced cell death in both WT and CMTM6-deficient HeLa cells (Fig. EV5F). Altogether, our data established that CMTM6 regulates mouse, but not human, FAS expression.

The strong interaction between CMTM6 and mouse FAS, mouse PD-L1, human PD-L1, and human CD58 prompted us to compare the amino acid sequences of their transmembrane and juxtamembrane domains. We noticed that the sequences are very different, with one notable exception: very strong conservation of arginine above the transmembrane domain (Fig. 7B), which our data demonstrated to be critical for the interaction of mouse FAS and CMTM6. Indeed, a detailed mutagenesis screen of amino acids at the boundary of the extracellular and transmembrane domains of

PD-L1 showed that mutation R238A or H240A nearly abolished the interaction with CMTM6 (Fig. 7C). Altogether, these data suggest that arginine positioned above the plasma membrane is required for CMTM6 interaction.

The analysis of the FAS transmembrane domain from various mammalian species shows that arginine above the transmembrane domain is very rare, as it is lacking even in rats. Interestingly, this residue is not present in the Asian lineage mouse *M. caroli*, but appears in the Palearctic mouse group, which includes *M. spretus, M. musculus*, and *M. spicilegus* (Fig. 7D). These mouse species diverged approximately 3–6 million years ago (Rudra et al, 2016), indicating that the evolution of FAS and CMTM6 interaction is a relatively recent evolutionary event.

## Discussion

In this work, we analyzed the interactome of six members of the mouse CMTM family, CMTM3-8. We did not include mouse CMTM1, CMTM2a, and CMTM2b as these proteins are unstable upon overexpression in ST2 cells (Knizkova et al, 2022). We also noticed that individual members of the CMTM family interacted with each other. This suggests that CMTM members may associate in the membrane, similarly to the proteins of the tetraspanin family, which also contain four transmembrane domains and short intracellular sequences, but differ in the presence of a large ectodomain. Tetraspanins create so-called tetraspanin nanodomains, which are important for the organization of membrane proteins and regulate diverse functions such as cell migration, cell signaling, or protein trafficking (Susa et al, 2024). However, whether the members of the CMTM family create functional nanodomains remains to be established.

CMTM6 is a widely studied protein due to its interaction with PD-L1 (Burr et al, 2017; Mezzadra et al, 2017). Our major focus was the identification of additional protein(s) that interact with mouse CMTM6 and could have a role in regulating tumor growth. We detected a relatively limited number of immunologically important transmembrane receptors. Most interestingly, we identified FAS as a strong interactor of mouse CMTM6, and we also weakly detected FAS in the mouse CMTM4 interactome, while it did not associate with other CMTM members. Our data indicated that CMTM6 binds to the transmembrane and juxtamembrane domain of FAS. PD-L1 employs a similar interaction mode to bind CMTM6 (Mezzadra et al, 2017), and therefore, it is possible that FAS competes with PD-L1 for CMTM6 binding.

CMTM4 and/or CMTM6 were shown to promote membrane localization of several proteins, such as PD-L1, CD58, IL-17RC, or EGFR (Burr et al, 2017; Knizkova et al, 2022; Mezzadra et al, 2017; Xu et al, 2025). Therefore, our initial assumption was that CMTM6 would also enhance FAS membrane localization. To our surprise, this was not the case, as naïve, memory, and effector T cells isolated from $Cmtm6^{-/-}$ mice had markedly enhanced membrane localization of FAS. In accord, $Cmtm6^{-/-}$ mice did not suffer from the lymphoproliferative disease that is typical for animals with impaired FAS expression or function (Watanabe-Fukunaga et al, 1992). Infection of $Cmtm6^{-/-}$ mice with L. monocytogenes induced normal formation of effector $CD8^+$ T cells, while the emergence of $CD4^+$ effector T cells was significantly decreased. In contrast to $CD8^+$ cells, $CD4^+$ effectors are highly sensitive to FAS-mediated activation-induced cell death (AICD), which functions as a mechanism of immune regulation (Zheng et al, 1995). Therefore, the decreased proportion of effector $CD4^+$ T cells upon infection in $Cmtm6^{-/-}$ mice compared to WT littermates is in accord with enhanced FAS expression and presumably increased AICD of these cells.

Mechanistically, CMTM6 facilitates the removal of FAS from the plasma membrane in mouse cells, and we noted strong colocalization of FAS and CMTM6 in recycling endosomes. Interestingly, CMTM4 was shown to promote the endocytosis of VE-cadherin and its internalization (Chrifi et al, 2019), suggesting that CMTM family members can have distinct functions depending on the receptor they are associated with. The role of CMTM6 as a negative regulator of FAS surface expression was further supported by the increased sensitivity of CMTM6-deficient mouse cells to FASL-induced cell death. However, in human cells, FAS is not associated with CMTM6 due to the difference in three amino acids at the extracellular and transmembrane domain boundary.

Mice are widely used animals to study tumor development. Since CMTM6 promotes PD-L1 expression, the role of CMTM6 in tumor progression was extensively studied. Ablation of CMTM6 decreased tumor growth in various mouse models due to increased T-cell cytotoxicity (Long et al, 2023). This effect was correlated with the decreased expression of immunosuppressive immune checkpoint PD-L1. Still, other mechanisms contributed to the increased cytotoxicity against CMTM6-negative tumors, as the combined deficiency of CMTM6 and PD-L1 led to a stronger anti-tumor immune reaction than the absence of either protein alone (Long et al, 2023). It is possible that CMTM6-deficient mouse tumor cells are prone to T-cell-mediated cytotoxicity due to enhanced plasma membrane expression of FAS. In support of this hypothesis, tumor cells can decrease FAS surface expression via enhanced endocytosis and lysosomal degradation. Slowing FAS endocytosis or promoting its recycling to the plasma membrane led to increased FAS surface expression and enhanced FASL-induced killing of target cells (Kural et al, 2024; Sharma et al, 2019). The dual role of CMTM6 in enhancing expression of immunosuppressive PD-L1 while suppressing expression of death-inducing FAS in mouse cells is an intriguing mechanism by which CMTM6 might prevent cytotoxic attack against self and might therefore play an important role in preventing anti-tumor immunity in mice.

In contrast, human FAS is not regulated by CMTM6. Instead, we noted a strong interaction between human CMTM6 and the costimulatory receptor CD58. This is in accord with previous reports showing that in human cells, CMTM6 is required for membrane expression of both PD-L1 and CD58 (Ho et al, 2023; Miao et al, 2023). CD58 is an adhesion molecule whose ligand, CD2, is expressed predominantly on T cells and NK cells. CD58 enables adhesion and costimulation of cytotoxic T cells and is required for anti-tumor immunity. This protein is, however, missing in rodents (Zhang et al, 2021). Since CMTM6-deficient human tumor cells have reduced CD58, they can escape anti-tumor immunity, and increased CMTM6 levels are associated with favorable immune checkpoint therapies in some cancers (Miao et al, 2023). Altogether, our results indicate that CMTM6 functions differently in human and mouse cells, as it regulates distinct sets of associated proteins beyond PD-L1. Therefore, the therapeutic potential of targeting human CMTM6 must be further evaluated, as mouse tumor models cannot fully recapitulate the function of human CMTM6 in anti-tumor immunity.

## Methods

**Reagents and tools table**

| Reagent/Resource | Reference or Source | Identifier or Catalog Number |
|---|---|---|
| **Experimental models** | | |
| ST2 cells (*M. musculus*) | Provided by Dr. Jana Balounova, IMG CAS, Czech Republic | |
| HeLa cells (*H. sapiens*) | Provided by Dr. Tomas Brdicka, IMG CAS, Czech Republic | |

| Reagent/Resource | Reference or Source | Identifier or Catalog Number |
|---|---|---|
| HEK293T cells (*H. sapiens*) | Provided by Dr. Tomas Brdicka, IMG CAS, Czech Republic | |
| Phoenix-Eco (*H. sapiens*) | Provided by Dr. Tomas Brdicka, IMG CAS, Czech Republic | |
| Phoenix-Ampho (*H. sapiens*) | Provided by Dr. Tomas Brdicka, IMG CAS, Czech Republic | |
| ST2 *Cmtm4* KO | This study | |
| ST2 *Fas* KO | This study | |
| HeLa *CMTM6* KO | This study | |
| HEK293T *FAS* KO | This study | |
| MEFs $Cmtm6^{+/+}$ | This study | |
| MEFs $Cmtm6^{-/-}$ | This study | |
| BMDMs $Cmtm6^{+/+}$ | This study | |
| BMDMs $Cmtm6^{-/-}$ | This study | |
| Mouse strain: C57BL/6N-A$^{tm1Brd}$ Cmtm6$^{tm1a(EUCOMM)Wtsi}$/WtsiCnbc | INFRAFRONTIER/EMMA | EM:06094 |
| Flp-deleter mouse strain, Gt(ROSA)26Sor$^{tm2(CAG-flpo,-EYFP)Ics}$ | IMG CAS, Czech Republic | MGI:5285396 |
| Cre-deleter strain, Gt(ROSA)26Sor$^{tm1(ACTB-cre,-EGFP)Ics}$ | IMG CAS, Czech Republic | MGI:5285392 |
| **Plasmids** | | |
| pBabe-Puro empty vector | Addgene | 21836 |
| pSpCas9(BB)-2A-GFP (PX458) | Addgene | 48138 |
| pBabe-EGFP | Provided by Dr. M. Hrdinka, University Hospital Ostrava, Czech Republic | |
| Mouse CMTM3-SF in pBabe | This study, GeneArt DNA synthesis | |
| Mouse CMTM4-SF in pBabe-EGFP | This study, GeneArt DNA synthesis | |
| Mouse CMTM5-SF in pBabe-EGFP | This study, GeneArt DNA synthesis | |
| Mouse CMTM6-SF in pBabe-EGFP | This study, GeneArt DNA synthesis | |
| Mouse CMTM7-SF in pBabe-EGFP | This study, GeneArt DNA synthesis | |
| Mouse CMTM8-SF in pBabe-EGFP | This study, GeneArt DNA synthesis | |
| Mouse CMTM6 in pBabe-EGFP | This study, GeneArt DNA synthesis | |
| Human CMTM6-SF in pBabe | This study, GeneArt DNA synthesis | |
| Human PD-L1-SF in pBabe-EGFP | This study, GeneArt DNA synthesis | |
| Human FAS-SF in pBabe-EGFP | This study, GeneArt DNA synthesis | |
| Mouse TRAILR2-SF | This study, GeneArt DNA synthesis | |

| Reagent/Resource | Reference or Source | Identifier or Catalog Number |
|---|---|---|
| Mouse FAS$^{Tm\_TRAILR2}$-SF | This study, GeneArt DNA synthesis | |
| Mouse TRAILR2$^{Tm\_FAS}$-SF | This study, GeneArt DNA synthesis | |
| Human FAS-SF | This study, GeneArt DNA synthesis | |
| Human FAS$^{Mut1}$-SF | This study, GeneArt DNA synthesis | |
| Human FAS$^{Mut2}$-SF | This study, GeneArt DNA synthesis | |
| Human FAS$^{Mut3}$-SF | This study, GeneArt DNA synthesis | |
| Human FAS$^{Mut4}$-SF | This study, GeneArt DNA synthesis | |
| Human FAS$^{Mut5}$-SF | This study, GeneArt DNA synthesis | |
| Human FAS$^{Mut6}$-SF | This study, GeneArt DNA synthesis | |
| Human FAS$^{Mut7}$-SF | This study, GeneArt DNA synthesis | |
| Human FAS$^{Mut8}$-SF | This study, GeneArt DNA synthesis | |
| Human FAS$^{Mut9}$-SF | This study, GeneArt DNA synthesis | |
| Human FAS$^{Mut10}$-SF | This study, GeneArt DNA synthesis | |
| Human FAS$^{Mut11}$-SF | This study, GeneArt DNA synthesis | |
| Human FAS$^{Mut12}$-SF | This study, GeneArt DNA synthesis | |
| Human FAS$^{Mut13}$-SF | This study, GeneArt DNA synthesis | |
| Human FAS$^{Mut14}$-SF | This study, GeneArt DNA synthesis | |
| Human FAS$^{Mut15}$-SF | This study, GeneArt DNA synthesis | |
| Human FAS$^{Mut16}$-SF | This study, GeneArt DNA synthesis | |
| Human FAS$^{Mut17}$-SF | This study, GeneArt DNA synthesis | |
| Human FAS-EGFP in pBabe-EGFP | This study, GeneArt DNA synthesis | |
| Mouse CMTM6-mCherry in pBabe-EGFP | This study, GeneArt DNA synthesis | |
| Human PD-L1_$A_{232}$G-Alfa in pBabe-Puro | This study, GeneArt DNA synthesis | |
| Human PD-L1_$H_{233}$A-Alfa in pBabe-Puro | This study, GeneArt DNA synthesis | |
| Human PD-L1_$P_{234}$A-Alfa-T2A-EGFP in pBabe-Puro | This study, GeneArt DNA synthesis | |
| Human PD-L1_$P_{235}$A-Alfa-T2A-EGFP in pBabe-Puro | This study, GeneArt DNA synthesis | |
| Human PD-L1_$N_{236}$A-Alfa-T2A-EGFP in pBabe-Puro | pBabe Puro, This study, GeneArt | |
| Human PD-L1_$E_{237}$A-Alfa-T2A-EGFP in pBabe-Puro | pBabe Puro, This study, GeneArt | |
| Human PD-L1_$R_{238}$A-Alfa-T2A-EGFP in pBabe-Puro | pBabe Puro, This study, GeneArt | |
| Human PD-L1_$T_{239}$A-Alfa-T2A-EGFP in pBabe-Puro | pBabe Puro, This study, GeneArt | |

| Reagent/Resource | Reference or Source | Identifier or Catalog Number |
|---|---|---|
| Human PD-L1_H$_{240}$A-Alfa-T2A-EGFP in pBabe-Puro | pBabe Puro, This study, GeneArt | |
| Human PD-L1_L$_{241}$A-Alfa-T2A-EGFP in pBabe-Puro | pBabe Puro, This study, GeneArt | |
| Human PD-L1_V$_{242}$A-Alfa-T2A-EGFP in pBabe-Puro | pBabe Puro, This study, GeneArt | |
| Human PD-L1_I$_{243}$A-Alfa in-T2A-EGFP pBabe-Puro | pBabe Puro, This study, GeneArt | |
| SP-His-SF-Hex-mFasL in pcDNA3.1 | pcDNA3.1, This study, GeneArt | |
| PX458 with sgRNA against mouse *Cmtm4* | This study | |
| PX458 with sgRNA against mouse *Fas* | This study | |
| PX458 with sgRNA against human *CMTM6* | This study | |
| PX458 with sgRNA against mouse *FAS* | This study | |
| **Antibodies** | | |
| Anti-β-actin | Cell Signaling Technology | 3700 |
| Anti-CMTM6 | Cell Signaling Technology | 90329 |
| Anti-caspase-8 | Cell Signaling Technology | 4790 |
| Anti-cleaved caspase-8 (Asp387) | Cell Signaling Technology | 9429 |
| Anti-cleaved caspase-3 (Asp175) | Cell Signaling Technology | 9664 |
| Anti-Flag | Cell Signaling Technology | 14793 |
| Anti-IL-17RC | R&D Systems | AF2270 |
| Anti-CMTM4 | Sigma-Aldrich | HPA014704 |
| Anti-Flag | Sigma-Aldrich | F3165 |
| Anti-mCherry | Thermofisher Scientific | M11217 |
| Anti-transferrin receptor | Thermofisher Scientific | 13-6800 |
| Anti-mouse FAS | Abcam | ab271016 |
| Anti-human FAS | Abcam | ab133619 |
| Anti-mouse FAS | Biolegend | 152602 |
| Anti-Human FAS | Biolegend | 305602 |
| Anti-Mouse IgG1, κ Isotype Ctrl | Biolegend | 401402 |
| Anti-B220 BV510 | Biolegend | 103248 |
| Anti-CD3 FITC | Biolegend | 100203 |
| Anti-CD4 AF700 | Biolegend | 100536 |
| Anti-CD8a BV421 | Biolegend | 100738 |
| Anti-CD44 PE | Biolegend | 103007 |
| Anti-CD44 PerCP-Cy5.5 | Biolegend | 103032 |

| Reagent/Resource | Reference or Source | Identifier or Catalog Number |
|---|---|---|
| Anti-CD49d AF647 | Biolegend | 103613 |
| Anti-CD62L BV510 | Biolegend | 104441 |
| Anti-CD62L PE-Cy7 | Biolegend | 104418 |
| Anti-mouse FASL APC | Biolegend | 106609 |
| Anti-mouse FAS FITC | Biolegend | 152605 |
| Anti-mouse FAS APC | Biolegend | 152604 |
| Anti-human FAS APC | Biolegend | 305611 |
| Anti-human PD-L1 APC | Biolegend | 329707 |
| Anti-human CD58 APC | Biolegend | 330917 |
| Anti-FoxP3 PE-Cy7 | Thermo Fisher Scientific | 25-5773-80 |
| Goat anti-rabbit IgG (H + L) AF488 | Thermo Fisher Scientific | A-11034 |
| Goat anti-mouse IgG AF647 | Thermo Fisher Scientific | A-21235 |
| Goat anti-rat IgG (H + L) AF594 | Thermo Fisher Scientific | A-11007 |
| Donkey anti-goat IgG (H + L) AF647 | Jackson Immunoresearch | 705-605-147 |
| Donkey anti-rabbit IgG (H + L) HRP | Jackson Immunoresearch | 711-035-152 |
| Goat anti-mouse IgG1 HRP | Jackson Immunoresearch | 115-035-205 |
| Goat anti-mouse IgG2b HRP | Jackson Immunoresearch | 115-035-207 |
| Donkey anti-rabbit IgG IRDye 800CW | LI-COR Bio | 926-32213 |
| Donkey anti-mouse IgG IRDye 800CW | LI-COR Bio | 926-32212 |
| Anti-Alfa tag IRDye 680RD | NanoTag Biotechnologies | N1502 |
| **Oligonucleotides and other sequence-based reagents** | **Forward sequence (5'-3')** | |
| Mouse *Cmtm4* sgRNA targeting sequence | GAAGTAGAGGCCTTCGCACG | |
| Mouse *Fas* sgRNA targeting sequence | GGCATGGTTGACAGCAAAAT | |
| Human *CMTM6* sgRNA targeting sequence | GTGAGAACGCGCCGGAGCAAT | |
| Human *FAS* sgRNA targeting sequence | GATCCAGATCTAACTTGGGG | |
| Genotyping mouse Cmtm4 WT allele - FWD primer | GCTGCTGTTTCTCATTGCTG | |
| Genotyping mouse Cmtm4 WT allele - REV primer | TGTGTCAAACGCTAAGACTCAGA | |
| Genotyping mouse Cmtm4 KO allele - FWD primer | GCTGCTGTTTCTCATTGCTG | |

| Reagent/Resource | Reference or Source | Identifier or Catalog Number |
|---|---|---|
| Genotyping mouse Cmtm4 KO allele - REV primer | GCTATGAACTGATGGCGAGC | |
| pBABE sequencing primer FWD | CCCCTTGAACCTCCTCTTTC | |
| pBABE sequencing primer REV | ACTTTCCACACCTGGTTGCT | |
| pcDNA3.1 sequencing primer FWD | CCACTGCTTACTGGCTTATCG | |
| pcDNA3.1 sequencing primer REV | CAACAGATGGCTGGCAACTA | |
| PX458 U6 sequencing primer FWD | GAGGGCCTATTTCCCATGATTCC | |
| **Cell culture and transfection** | | |
| Dulbecco's modified Eagle medium, high glucose (DMEM) | Sigma-Aldrich | #D6429 |
| Fetal Bovine Serum (FBS) | Biosera | #FB-1001H |
| Penicilin/ streptomycin antibiotics | Biosera | #XC-A4122 |
| Non-essential amino acids | Sigma-Aldrich | M7145 |
| M-CSF | PeproTech | 315-02-10UG |
| Trypsin-EDTA | Biosera | XC-T1717 |
| Lipofectamin 2000 | Invitrogen | 11668-027 |
| Polyethylenimin (PEI) | Polysciences | 23966-2 |
| Polybrene | Sigma-Aldrich | TR-1003 |
| **Chemicals, Enzymes and other reagents** | | |
| T4 DNA Ligase | ThermoFisher Scientific | EL0016 |
| AgeI restriction enzyme | ThermoFisher Scientific | FD1464 |
| EcoRI restriction enzyme | ThermoFisher Scientific | FD0274 |
| BglII restriction enzyme | ThermoFisher Scientific | FD0083 |
| Zymoclean Gel DNA Recovery Kit | Zymo Research | D4001 |
| Plasmid Mini Kit II | E.Z.N.A., Omega Bio-Tek | D6945-02 |
| n-Dodecyl-beta-D-Maltoside | ThermoFisher Scientific | 89903 |
| PhosSTOP Phosphatase Inhibitor Tablets | Roche | 04906837001 |
| cOmplete, EDTA-free Protease Inhibitor Tablets | Roche | 05056489001 |
| Anti-Flag M2 affinity agarose gel | Sigma-Aldrich | A2220 |
| 3xFlag peptide | Sigma-Aldrich | F4799 |
| Strep-Tactin sepharose beads | IBA Lifesciences | 2-1201-010 |

| Reagent/Resource | Reference or Source | Identifier or Catalog Number |
|---|---|---|
| Alfa Selector magnetic resin | Nano Tag Biotechnologies | N1511 |
| Protein A/G PLUS agarose | Santa Cruz Biotechnology | sc-2003 |
| eTox Red Dye | Agilent | 8711009 |
| LIVE/DEAD near-IR dye | Thermo Fisher Scientific | L34976 |
| Foxp3 transcription factor staining buffer set | Thermo Fisher Scientific | 00-5523-00 |
| Chameleon® Duo Pre-stained Protein Ladder | LI-COR Bio | 928-60000 |
| Precision Plus Protein Dual Color Standards | Bio-Rad | 1610374 |
| Human Hex-FASL | AdipoGen | AG-40B-0130-C010 |
| Prolong Gold reagent | Thermo Fisher Scientific | P36930 |
| Taq polymerase (Combi PPP Master Mix) | Top-Bio | C210 |
| **Software** | | |
| MaxQuant software 2.4.13.0 | Cox et al, 2014 | |
| Perseus 1.6.14.0 | Tyanova et al, 2016 | |
| GraphPad Prism 8.0 | https://www.graphpad.com/ | |
| ImageLab | Bio-Rad | |
| FlowJo Software | TreeStar | |
| Zen | ZEISS | |
| **Other** | | |
| CHOPCHOP | Labun et al, 2019 | |
| Zeiss LSM 880 confocal microscope | ZEISS | |
| Nanodrop 2000 spectrophotometer | Thermo Fisher Scientific | |
| ChemiDoc MP | Bio-Rad | |
| Incucyte SX1 | Sartorius | |
| Bricyte E6 flow cytometer | Mindray | |
| CytoFlex flow cytometer | Beckman Coulter | |
| Aurora flow cytometer | Cytek | |
| FACSAria III cell sorter | BD Bioscience | |
| Odyssey CLx imaging system | LI-COR Biosciences | |

## Antibodies

For immunoblotting, the primary antibodies used were anti-β-actin (catalog number #3700), anti-CMTM6 (#90329), anti-Flag (#14793), anti-cleaved caspase-8 (Asp387) (#9429), anti-cleaved

caspase-3 (Asp175) (#9664), and anti-caspase-8 (#4790) from Cell Signaling Technology, anti-CMTM4 (#HPA014704) and anti-Flag (#F3165) from Sigma-Aldrich, anti-mouse FAS (#ab271016) and anti-human FAS (#ab133619) from Abcam. Secondary antibodies used were donkey anti-rabbit IgG (H + L) HRP (#711-035-152), goat anti-mouse IgG1 HRP (#115-035-205), and goat anti-mouse IgG2b HRP (#115-035-207) from Jackson Immunoresearch. For the analysis of immunoblots using fluorescence, we used anti-Alfa tag IRDye 680RD (#N1502) from NanoTag Biotechnologies, donkey anti-rabbit IgG IRDye 800CW (#926-32213), and donkey anti-mouse IgG IRDye 800CW (#926-32212) from LI-COR Biosciences.

For flow cytometry, we used following fluorescently labeled antibodies anti-B220 BV510 (#103248), anti-CD3 FITC (#100203), anti-CD4 AF700 (#100536), anti-CD8a BV421 (#100738), anti-CD44 PE (#103007), anti-CD44 PerCP-Cy5.5 (#103032), anti-CD49d AF647 (#103613), anti-CD62L BV510 (#104441), anti-CD62L PE-Cy7 (#104418), anti-FASL APC (#106609), anti-mouse FAS FITC (#152605), anti-mouse FAS APC (#152604), anti-human FAS APC (#305611), anti-human PD-L1 APC (#329707), and anti-human CD58 APC (#330917) from BioLegend and FoxP3 PE-Cy7 (#25-5773-80) from Thermo Fisher Scientific. Unconjugated Anti-IL-17RC (#AF2270) from R&D Systems was detected with donkey anti-goat IgG (H + L) AF647 (#705-605-147) from Jackson Immunoresearch.

For microscopy, we employed the following antibodies: anti-FAS (#ab271016) from Abcam, anti-mCherry (#M11217), anti-transferrin receptor (#13-6800), and secondary antibodies goat anti-rabbit IgG (H + L) AF488 (#A-11034), goat anti-mouse IgG AF647 (#A-21235), and goat anti-rat IgG (H + L) AF594 (#A-11007) from Thermo Fisher Scientific.

For endogenous FAS immunoprecipitation and internalization assays, we employed anti-mouse FAS (#152602), anti-human FAS (#305602), or an IgG1 isotype control (#401402) antibodies from BioLegend.

## Recombinant proteins

Recombinant mouse Hex-FASL consisted of CD33 leader, 6xHis tag, 2xStrep-1xFlag tag, oligomerization domain from mouse adiponectin (AA 18-111) (Holler et al, 2003), and C-terminal part of mouse FASL (AA 136-279). Recombinant human IFNγ consisted of CD33 leader, 6xHis tag, 2xStrep-1xFlag tag, and IFNγ (AA 24-166). To produce each recombinant protein, the DNA coding sequence was synthesized using a GeneArt Gene Synthesis service (Thermo Fisher Scientific) and cloned into the pcDNA3.1 vector. The resulting constructs were transfected into HEK293T for using polyethylenimine (PEI) transfection. To prevent the killing of cells by newly produced Hex-FASL, we used HEK293T FAS$^{KO}$ cells. After three days, the culture supernatant was collected and Hex-FASL was purified using a His-GraviTrap TALON column (GE Healthcare) equilibrated with purification buffer (50 mM sodium phosphate, pH 7.4, 300 mM NaCl). The column was washed with purification buffer containing 20 mM imidazole, and Hex-FASL was eluted with purification buffer containing 350 mM imidazole. Following elution, imidazole was removed, and the buffer was exchanged using an Amicon Ultra centrifugal filter with a 10 kDa molecular weight cutoff (Merck). Samples were concentrated by centrifugation and washed three times with purification buffer. The concentration of recombinant protein was measured on a Nanodrop 2000 spectrophotometer (Thermo Fisher Scientific). The purified protein solution was mixed with glycerol to a final concentration of 50% and stored at −80 °C for long-term preservation. The purity of the recombinant ligand was assessed by SDS-PAGE followed by staining with Coomassie InstantBlue (Expedeon). Recombinant human Hex-FASL was purchased from AdipoGen.

## Cell lines and animal models

ST2 cells were kindly provided by J. Balounova, and HeLa, HEK293T, Phoenix-Eco, and Phoenix-Ampho cells were kindly provided by T. Brdicka (both from the Institute of Molecular Genetics, Prague, Czech Republic). MEFs were isolated from E11.5 mouse embryos and immortalized via lentiviral transduction with the SV40 large T antigen. All cell lines were cultured at 37 °C in a humidified atmosphere with 5% CO$_2$ in complete Dulbecco's Modified Eagle Medium (DMEM), supplemented with 10% fetal bovine serum (FBS) (Biosera) and penicillin/streptomycin antibiotics (Biosera). To obtain BMDMs, bone marrow cells from the femur and tibia of 10-week-old mice were differentiated in DMEM containing 10% FBS, penicillin/streptomycin, 1% non-essential amino acids (Sigma-Aldrich), and 25 ng/ml of M-CSF (PeproTech) for 7 days. Cell lines were regularly tested for Mycoplasma contamination using the Mycoplasmacheck service (Eurofins Genomics).

Frozen sperm from mouse strain C57BL/6N-A$^{tm1Brd}$ Cmtm6$^{tm1a(EUCOMM)Wtsi}$/WtsiCnbc was provided by The Wellcome Trust Sanger Institute and obtained via the INFRAFRONTIER/ EMMA repository (EM:06094) (Consortium, 2015). Upon in vitro fertilization, mice carrying the targeted Cmtm6 allele were crossed to the Flp-deleter mouse strain, Gt(ROSA)26Sor$^{tm2(CAG-flpo,-EYFP)Ics}$ (MGI:5285396). The resulting mice, with exon 2 flanked by LoxP sites, were then crossed with the Cre-deleter strain, Gt(ROSA) 26Sor$^{tm1(ACTB-cre,-EGFP)Ics}$ (MGI:5285392), to obtain mice carrying the Cmtm6 KO allele. Cmtm6$^{+/+}$ and Cmtm6$^{−/−}$ male and female littermates used in experiments were generated by breeding heterozygous animals. Mice were housed under specific pathogen-free conditions, with a 12-h light/12-h dark cycle, a temperature of 22 ± 1 °C, and a relative humidity of 55 ± 5%. Genotyping was performed using PCR with Taq polymerase (Top-Bio) and the primer pairs for Cmtm6 wild-type allele (5'-GCTGCTGTTT CTCATTGCTG-3', 5'-TGTGTCAAACGCTAAGACTCAGA-3') and KO allele (5'-GCTGCTGTTTCTCATTGCTG-3', 5'-GCTATG AACTGATGGCGAGC-3'). Animal protocols were approved by the Czech Academy of Sciences, Czech Republic (AVCR 5262/2024 SOV II).

## Generation of KO cell lines

KO cell lines were generated using the CRISPR-Cas9 approach. Single-guided RNAs (sgRNA) were designed using a CHOPCHOP web tool (Labun et al, 2019). We used sgRNA targeting mouse Cmtm4 (5'-GAAGTAGAGGCCTTCGCACG-3') and mouse Fas (5'-GGCATGGTTGACAGCAAAAT-3') in ST2 cells, human CMTM6 (5'-GTGAGAACGCGCCGGAGCAAT-3') in HeLa cells, and human FAS (5'-GATCCAGATCTAACTTGGGG-3') in HEK293T. The sgRNA sequences were cloned into pSpCas9(BB)-2A-GFP (PX458) vector provided by Feng Zhang (Addgene plasmid #48138) (Ran et al, 2013), and transfected into cell lines using Lipofectamine 2000 (Thermo Fisher Scientific). GFP-positive cells were isolated using a FACSAria III cell sorter (BD Bioscience) and subcloned. Target protein expression was assessed by immunoblotting.

## DNA cloning and viral transduction

Coding sequences of indicated proteins were synthesized using the GeneArt Gene Synthesis service (Thermo Fisher Scientific) and cloned into the retroviral pBabe vector expressing EGFP selection marker under the SV40 promoter (kindly provided by M. Hrdinka, University Hospital Ostrava, Czech Republic). Transmembrane and juxtamembrane domains of mouse FAS (AA165-193) and mouse TRAILR2 (AA175-209) were swapped to map interaction with CMTM6. The coding sequence of PD-L1-Alfa, in frame with T2A-EGFP, was synthesized and cloned into the pBabe Puro vector, provided by David Ron (Addgene plasmid #21836).

Phoenix-Eco cells (for mouse cell transduction) or Phoenix-Ampho cells (for human cell transduction) were transfected with the retroviral vectors using Lipofectamine 2000 (Thermo Fisher Scientific) to produce retroviral particles. Virus-containing supernatants were collected, filtered through a 0.2 μm filter, and used to transduce target mouse (ST2, MEFs) or human (HeLa) cell lines in the presence of polybrene (6 μg/ml, Sigma-Aldrich). Transduction was facilitated by spinfection (2500 rpm, 45 min, 30 °C). Successfully transduced cells were sorted for EGFP expression using a FACSAria IIu cell sorter (BD Biosciences).

## Cell stimulation, cell lysis, immunoprecipitation, and immunoblotting

For the analysis of FASL-induced caspase-3 and caspase-8 activation, cells were washed with serum-free DMEM and stimulated with 500 ng/ml of mouse Hex-FASL. After stimulation, cells were solubilized on ice for 30 min in lysis buffer (30 mM Tris, pH 7.4, 120 mM NaCl, 2 mM KCl, 2 mM EDTA, 10% glycerol, 10 mM chloroacetamide, Complete protease inhibitor cocktail, and PhosSTOP tablets (Roche)) containing 1% n-Dodecyl-β-D-Maltoside (DDM) (Thermo Fisher Scientific). Lysates were cleared by centrifugation ($21,130 \times g$, 30 min, 2 °C). Cleared lysates were mixed with 4x SDS sample buffer (250 mM Tris pH 6.8, 8% SDS, 40% glycerol, 0.2% bromophenol blue), reduced with 50 mM dithiothreitol (DTT), and heated (92 °C, 3 min).

For the isolation of proteins via immunoprecipitation, cells were washed with DMEM, solubilized in lysis buffer containing 1% DDM on ice for 30 min, and cleared by centrifugation (21,130 g, 30 min, 2 °C). A portion of the cleared lysates was mixed with 4x SDS sample buffer, reduced with 50 mM DTT, and heated (92 °C, 3 min) to prepare the lysate samples. To isolate exogenously expressed SF-tagged or Alfa-tagged proteins, lysates were subjected to overnight immunoprecipitation using either anti-Flag M2 affinity agarose gel (Sigma-Aldrich) or Alfa Selector ST magnetic resin (NanoTag Biotechnologies). To analyze FAS interacting partners, cellular lysates were incubated with anti-FAS or control antibody for 1 h at 4 °C, followed by overnight immunoprecipitation with Protein A/G PLUS agarose (Santa Cruz Biotechnology). The next day, beads were washed three times with lysis buffer containing 0.1% DDM and subsequently mixed with 4x SDS sample buffer, reduced with 50 mM DTT, and heated (92 °C, 3 min).

Samples were analyzed via immunoblotting using the indicated antibodies. Secondary antibodies were either conjugated to HRP and detected using the ChemiDoc MP imaging system (Bio-Rad) (Figs. 1–6) or to fluorescence dye and visualized using the Odyssey CLx imaging system (LI-COR Biosciences) (Fig. 7).

## Tandem affinity purification and mass spectrometry analysis

For each experimental condition, murine ST2 or human HeLa cells expressing the indicated proteins were cultured on six 15-cm dishes. Cells were washed with serum-free DMEM and lysed on ice for 30 min in lysis buffer containing 1% DDM. Lysates were cleared by centrifugation ($21,130 \times g$, 30 min, 2 °C) followed by a two-step immunoprecipitation.

In the first step, lysates were incubated overnight with 50 μl of anti-Flag M2 affinity agarose gel (Sigma-Aldrich). The next day, the beads were washed three times with lysis buffer containing 0.1% DDM. Bound proteins were eluted overnight in 250 μl of lysis buffer containing 1% DDM and 100 μg/ml 3xFlag peptide (Sigma-Aldrich). A second elution performed for an additional 6 h, and both eluates were pooled. The second purification step was overnight incubation of the pooled supernatants with 50 μl of Strep-Tactin Sepharose beads (IBA Lifesciences). Samples were subsequently washed three times with lysis buffer containing 0.1% DDM and once with lysis buffer alone. Finally, bound proteins were eluted by incubating the beads with 220 μl of elution buffer (2% sodium deoxycholate, 50 mM Tris, pH 8.5).

The eluted protein samples (200 μl) were reduced with 5 mM tris(2-carboxyethyl)phosphine at 60 °C for 60 min and alkylated with 10 mM methyl methanethiosulfonate at room temperature for 10 min. Proteins were cleaved overnight with 1 μg of trypsin (Promega) at 37 °C. To remove sodium deoxycholate, samples were acidified with 1% trifluoroacetic acid, mixed with an equal volume of ethyl acetate, centrifuged ($15,700 \times g$, 2 min), and an aqueous phase containing peptides was collected (Masuda et al, 2008). This extraction step was repeated two more times. Peptides were desalted using in-house-made stage tips packed with C18 disks (Empore) (Rappsilber et al, 2007) and resuspended in 20 μl of 2% acetonitrile with 1% trifluoroacetic acid.

The digested protein samples (12 μl) were loaded onto the trap column (Acclaim PepMap300, C18, 5 μm, 300 Å Wide Pore, 300 μm × 5 mm) using 2% acetonitrile with 0.1% trifluoroacetic acid at a flow rate of 15 μl/min for 4 min. Subsequently, peptides were separated on a Nano Reversed-phase column (EASY-Spray column, 50 cm × 75 μm internal diameter, packed with PepMap C18, 2 μm particles, 100 Å pore size) using a linear gradient from 4% to 35% acetonitrile containing 0.1% formic acid at a flow rate of 300 nl min$^{-1}$ for 60 min.

Ionized peptides were analyzed on a Thermo Orbitrap Fusion (Q-OT-qIT, Thermo Fisher Scientific). Survey scans of peptide precursors from 350 to 1400 m/z were performed at 120 K resolution setting with a $4 \times 10^5$ ion count target. Four different types of tandem mass spectrometry were performed according to precursor intensity. The first three types were detected in Ion trap in rapid mode, and the last one was detected in Orbitrap with 15,000 resolution settings: (1) For precursors with intensity between $1 \times 10^3$ to $7 \times 10^3$ with collision-induced dissociation (CID) fragmentation (35% collision energy) and 250 ms of ion injection time; (2) For ions with intensity in range from $7 \times 10^3$ to $9 \times 10^4$ with CID fragmentation (35% collision energy) and 100 ms of ion injection time; (3) For ions with intensity in range from $9 \times 10^4$ to $5 \times 10^6$ with higher-energy-collisional-dissociation (HCD) fragmentation (30% collision energy) and 100 ms of ion injection time; and (4) For intensities $5 \times 10^6$ and more with HCD fragmentation (30% collision energy) and 35 ms of

ion injection time. The dynamic exclusion duration was set to 60 s with a 10 ppm tolerance around the selected precursor and its isotopes. Monoisotopic precursor selection was turned on. The instrument was run in top speed mode with 3 s cycles.

Mass spectrometry data were analyzed and quantified with the MaxQuant software (version 2.4.13.0) (Cox et al, 2014). The false discovery rate (FDR) was set to 1% for both proteins and peptides with a minimum peptide length of seven amino acids. The Andromeda search engine was used to search mass spectrometry data against the murine or human Swiss-Prot database (downloaded from Uniprot in October 2024). Trypsin specificity was set as C-terminal to Arg and Lys, also allowing the cleavage at proline bonds and a maximum of two missed cleavages. N-terminal protein acetylation, carbamidomethylation, and Met oxidation were included as variable modifications. Label-free quantification was performed using the Top3 algorithm, which uses the average intensity of the three most intense peptides per protein. Data analysis was performed using Perseus 1.6.14.0 software (Tyanova et al, 2016). To exclude common contaminants, we compared the list of potential interactors with the CRAPome database (Mellacheruvu et al, 2013).

## Flow cytometry of cell lines and primary cells

For the analysis of surface expression of indicated proteins in cell lines, cells were either left unstimulated or stimulated with IFN-γ (100 ng/ml) for 24 h to induce PD-L1 expression. Cells were resuspended in FACS buffer (PBS, 2% FBS, 0.1% NaN3), stained using fluorescently labeled antibodies for 30 min on ice. Cells were washed with ice-cold FACS buffer. Surface expression of selected proteins was measured on the Bricyte E6 flow cytometer (Mindray).

For the analysis of mouse immune cell populations, spleens and peripheral lymph nodes were collected from euthanized 8–12-week-old WT and $Cmtm6^{-/-}$ littermate mice. The organs were dissociated into single-cell suspensions using nylon mesh. Splenic cells were incubated in ACK lysis buffer (150 mM NH$_4$Cl, 10 mM KHCO$_3$, 0.1 mM EDTA-Na$_2$, pH 7.4) for 2 min at room temperature to remove red blood cells. Cells were resuspended in FACS buffer and stained on ice with the live/dead near-IR dye (Invitrogen), followed by fixation with Foxp3 transcription factor staining buffer set (Thermo Fisher Scientific). Fixed cells were washed, resuspended in FACS buffer, and stained on ice for 30 min with a mixture of anti-mouse antibodies consisting of anti-CD3 FITC, anti-CD4 AF700, anti-CD8a BV421, anti-CD44 PE, anti-CD49d AF647, anti-CD62L BV510, anti-FoxP3 PE-Cy7, and anti-FAS APC. Samples were washed and measured on an Aurora flow cytometer (Cytek). The flow cytometry data were analyzed by FlowJo Software (TreeStar).

## Mouse infections with *L. monocytogenes*

WT and $Cmtm6^{-/-}$ male and female littermates (7–8 weeks old) were infected intravenously with *L. monocytogenes* expressing OVA (5000 CFU in 200 μL PBS). After 10 days, mice were euthanized, and their spleens were collected and dissociated into single-cell suspensions using a nylon mesh. Red blood cells were lysed by incubation in ACK buffer for 2 min at room temperature. Cells were washed, resuspended in FACS buffer, and stained on ice for 30 min with an anti-mouse antibody mixture that included live/dead fixable near-IR dead stain (Invitrogen), anti-B220 BV510,

anti-CD4 AF700, anti-CD8a BV421, anti-CD44 PerCP-Cy5.5, anti-CD62L PE-Cy7, anti-FAS FITC, and anti-FASL. To detect antigen-specific cells, Kb-OVA tetramers were used, prepared by incubating biotinylated pMHC I monomers (NIH Tetramer Core Facility) with PE-streptavidin (Invitrogen). Samples were washed and analyzed using an Aurora flow cytometer (Cytek). Data were analyzed with FlowJo software (TreeStar).

## Analysis of cell death induction

Cells were seeded in 96-well plates in DMEM supplemented with 10% FBS and ATB at a density of 8000 cells per well. The following day, cells were stimulated or not with recombinant Hex-FASL (500 ng/ml) in the presence of 250 nM eTox Red Dye (Agilent) or Cytotox Red Dye (Sartorius) to visualize dead cells. The cells were monitored every 2 h over a 24-h period using an Incucyte SX1 live-cell analysis system (Sartorius). In each experiment, samples were analyzed in triplicate, and four images were captured per well. Cell death induction was quantified as the ratio of red fluorescent area to total cell confluency.

## Analysis of FAS internalization

For the analysis of FAS internalization, MEFs were resuspended in ice-cold DMEM supplemented with 10% FBS and stained with FAS antibody for 30 min on ice or left unstained. Cells were washed and resuspended in warm 1.5 ml of DMEM with 10% FBS and incubated at 37 °C for the indicated time points or left on ice as a control. After incubation, the cells were stained with A647-conjugated secondary antibody for 30 min on ice in the dark. Subsequently, cells were washed with ice-cold FACS buffer (PBS, 2% FBS, 0.1% NaN3). Surface expression of remaining FAS was measured on the CytoFlex flow cytometer (Beckman Coulter).

## Fluorescence microscopy

$Cmtm6^{-/-}$ MEFs expressing CMTM6-mCherry and FAS-EGFP were grown on round coverslips overnight. Slides were fixed using 4% formaldehyde in PBS for 10 min, permeabilized with 0.1% Triton X-100 in PBS for 10 min, and blocked with 5% goat serum in PBS for 1 h. Between each step, samples were washed three times with PBS-T (0.1% Tween-20 in PBS) for 5 min. Cells were stained with primary antibodies diluted in 1% goat serum in PBS overnight at 4 °C or for 1 h at room temperature. Subsequently, the samples were washed three times with PBS-T and once with PBS, and then stained with secondary antibodies diluted in 1% goat serum in PBS for 1 h. Cells were then washed three times with PBS-T, stained with 300 nM DAPI in PBS for 15 min, and finally washed stepwise with PBS-T, PBS, and H$_2$O for 5 min each. Dry slides were mounted in Prolong Gold reagent (Thermo Fisher Scientific) and analyzed using a Zeiss LSM 880 confocal microscope. Data were analyzed in Zen software (ZEISS).

## Statistics

The indicated statistical tests were performed using Prism (GraphPad Software). In the case of one-way ANOVA and an unpaired two-tailed t-test, the normality was assessed using the Shapiro-Wilk test.

# Data availability

The mass spectrometry proteomics data have been deposited to the ProteomeXchange Consortium via the PRIDE partner repository (Perez-Riverol et al, 2025). The dataset identifiers are PXD070246 for analysis of the mouse CMTM3-8 interactome, PXD070298 for the mouse FAS interactome, and PXD070359 for the human CMTM6 interactome.

The source data of this paper are collected in the following database record: biostudies:S-SCDT-10_1038-S44319-026-00694-8.

# Peer review information

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

## Acknowledgements

We thank J. Stefanovic for technical assistance. Mass spectrometry analysis was performed at the OMICS Mass Spectrometry Core Facility, Faculty of Science, Charles University, located at the BIOCEV research center. This project was supported by grants from the Czech Science Foundation (21-25251S) and the Pierre Mercier Foundation (26070176) awarded to PD. OS was supported by the National Institute of Virology and Bacteriology (Programme EXCELES, LX22NPO5103), funded by the European Union-Next Generation EU. TS and MP were students at the First Faculty of Medicine at Charles University in Prague and supported by a grant from the Ministry of Education, Youth and Sports of the Czech Republic (SVV 260763). VC was a student at the Faculty of Science at Charles University in Prague and was supported by a Charles University grant (176624).

## Author contributions

**Tereza Semberova**: Conceptualization; Formal analysis; Supervision; Investigation; Methodology; Writing—original draft; Writing—review and editing. **Michaela Pribikova**: Investigation; Methodology. **Veronika Cimermanova**: Formal analysis; Investigation; Methodology. **Tijana Trivic**: Formal analysis; Supervision; Investigation; Methodology. **Rafik Haderbache**: Formal analysis; Investigation; Methodology. **Darina Paprckova**: Formal analysis; Investigation; Methodology. **Luca Christen**: Investigation. **Helena Kissiova**: Investigation. **Ondrej Stepanek**: Resources; Supervision; Funding acquisition; Methodology; Project administration. **Peter Draber**: Conceptualization; Resources; Data curation; Formal analysis; Supervision; Funding acquisition; Visualization; Methodology; Writing—original draft; Project administration; Writing—review and editing.

Source data underlying figure panels in this paper may have individual authorship assigned. Where available, figure panel/source data authorship is listed in the following database record: biostudies:S-SCDT-10_1038-S44319-026-00694-8.

## Disclosure and competing interests statement

The authors declare no competing interests.

# Expanded View Figures

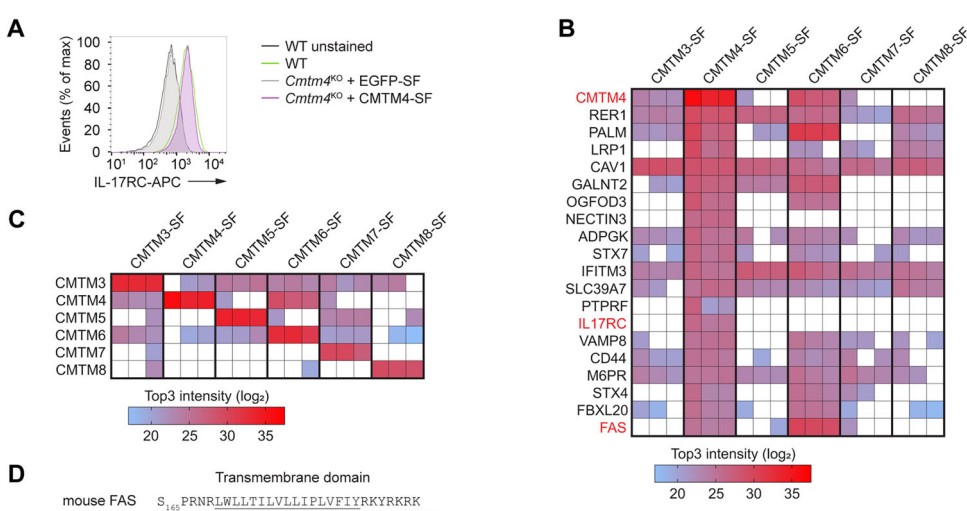

**Figure EV1. Analysis of the CMTM family interactome.**

(**A**) Flow cytometry analysis of surface IL-17RC expression in ST2 WT or *Cmtm4*[KO] cells transduced with expression vectors encoding EGFP-SF or CMTM4-SF. (**B**) Mass spectrometry analysis of the indicated Strep-Flag (SF)-tagged CMTM family members isolated from mouse ST2 cells via tandem affinity purification. The heatmap displays the most abundant CMTM4-associated proteins that were not detected in EGFP-SF control samples, based on Top3 intensities from 3 independent experiments. (**C**) Heatmap showing Top3 intensity values for the associations between various CMTM family members. (**D**) Sequence comparison of murine FAS and TRAILR2 transmembrane (underlined) and juxtamembrane domains. Source data are available online for this figure.

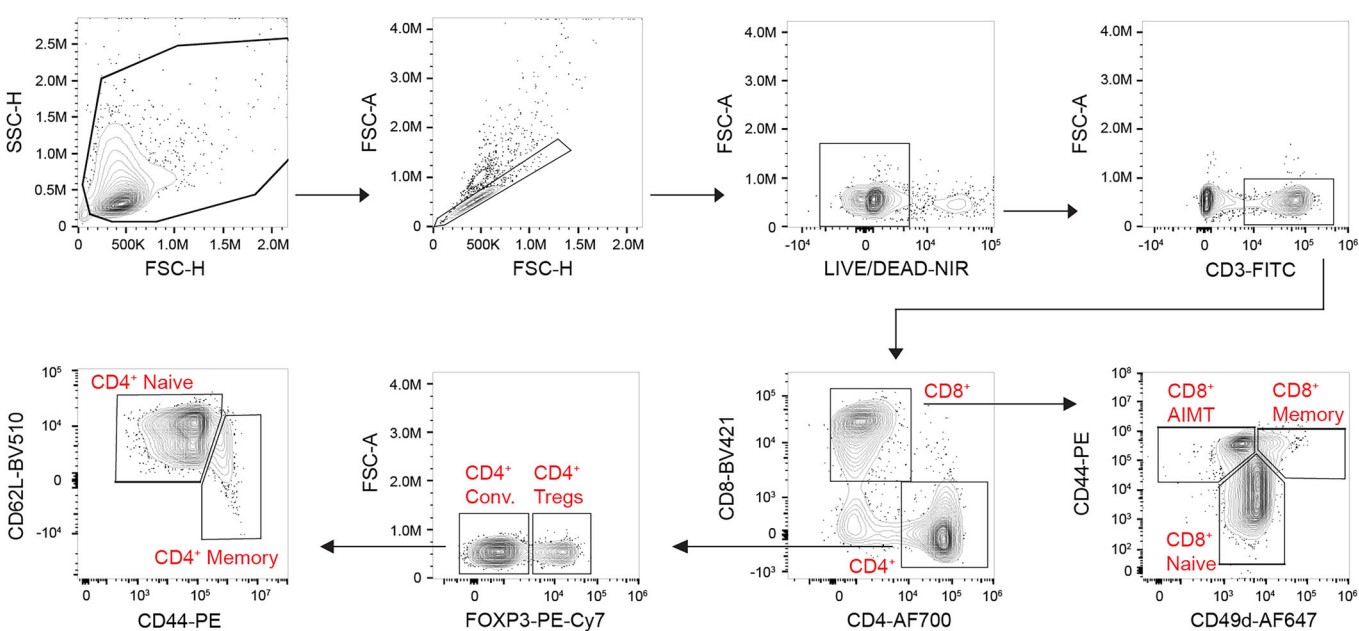

**Figure EV2. Mouse T cells gating strategy.**

The gating strategy used for analyzing T cells isolated from the spleen (Fig. 3) and peripheral lymph nodes (Fig. EV3).

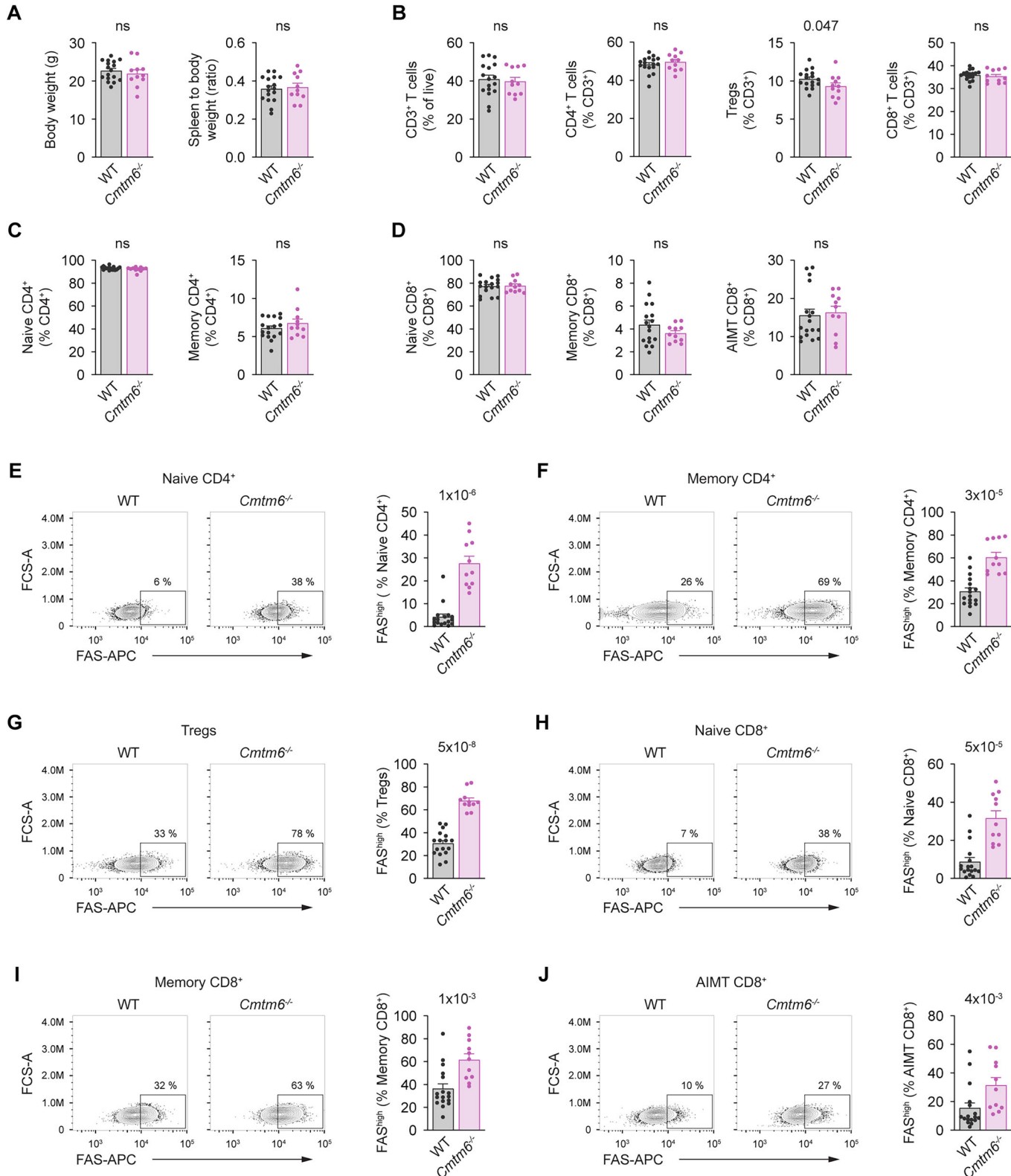

◀ **Figure EV3. CMTM6 suppresses FAS expression in T cells isolated from mouse lymph nodes.**

(A) Body weight and spleen-to-body weight ratio in 8-12-week-old WT and $Cmtm6^{-/-}$ mice. (B) Flow cytometry analysis of T cells isolated from peripheral lymph nodes of 8-12-week-old WT and $Cmtm6^{-/-}$ mice. T cells (gated as CD3$^+$) were subdivided into conventional CD4$^+$ (CD4$^+$, FOXP3$^-$), Tregs (CD4$^+$, FOXP3$^+$), and conventional CD8$^+$ (CD8$^+$). (C) Conventional CD4$^+$ T cells were further gated as naïve (CD44$^-$, CD62L$^+$) and memory (CD44$^+$, CD62L$^-$). (D) Conventional CD8$^+$ T cells were gated as naïve (CD44$^-$), memory (CD44$^+$, CD49d$^+$), and antigen-inexperienced memory T (AIMT) (CD44$^+$, CD49d$^-$) cells. (E–J) FAS expression in the indicated subsets of T cells isolated from peripheral lymph nodes. Data are presented as mean + SEM, $n = 17$ WT and 11 $Cmtm6^{-/-}$ mice. Two-tailed Mann-Whitney test. ns not significant. Source data are available online for this figure.

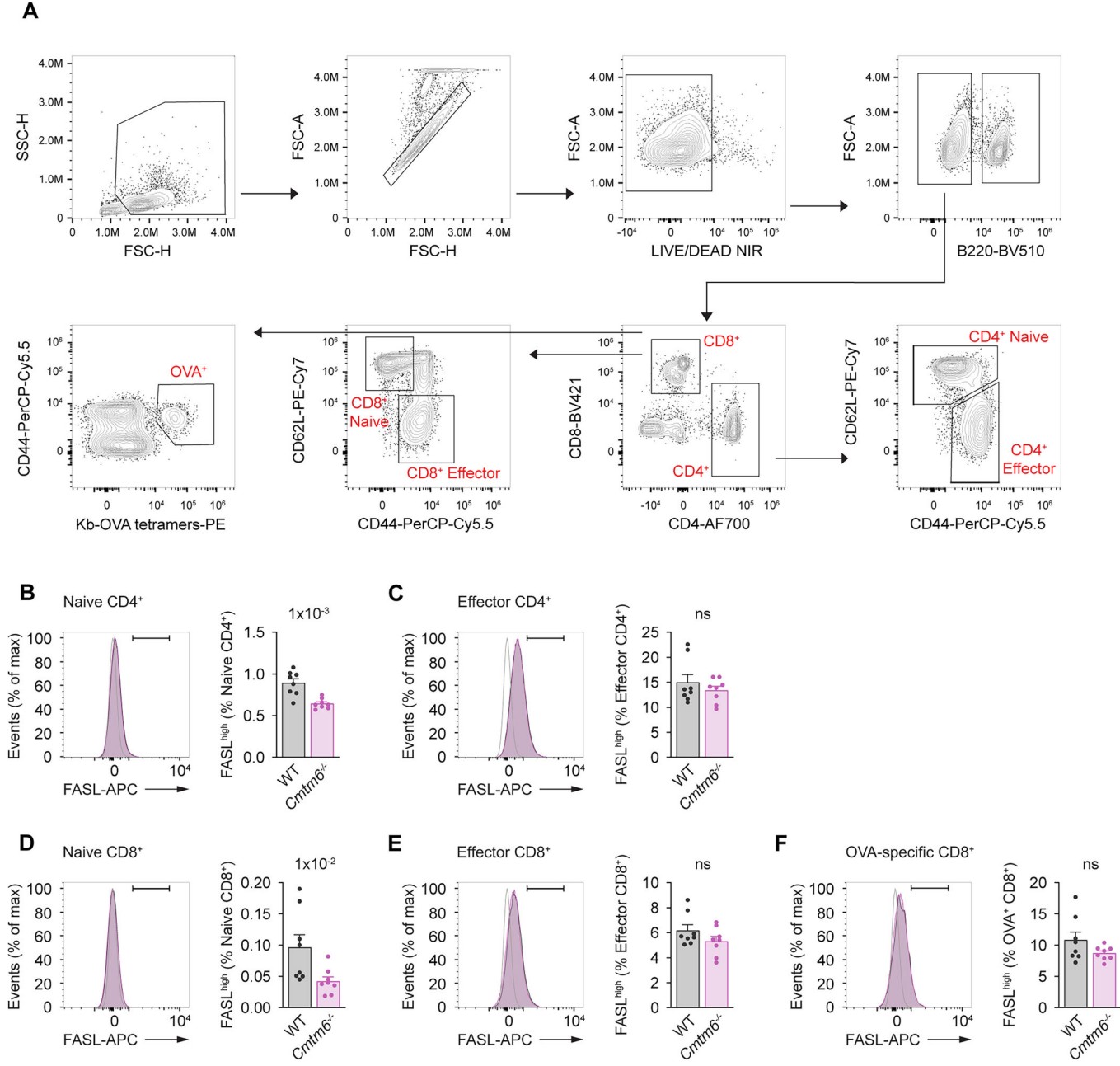

**Figure EV4. Role of CMTM6 in infection with *L. Monocytogenes*.**

(A) The gating strategy used for the analysis of T cells isolated from the spleens of mice 10 days post-infection with *L. Monocytogenes* (Figs. 4 and EV4). (B–F) Surface FASL levels in the indicated subsets of splenic T cells. Data are presented as mean + SEM from two independent experiments. *n* = 8 mice per group. Two-tailed Mann-Whitney statistical test. ns not significant. Source data are available online for this figure.

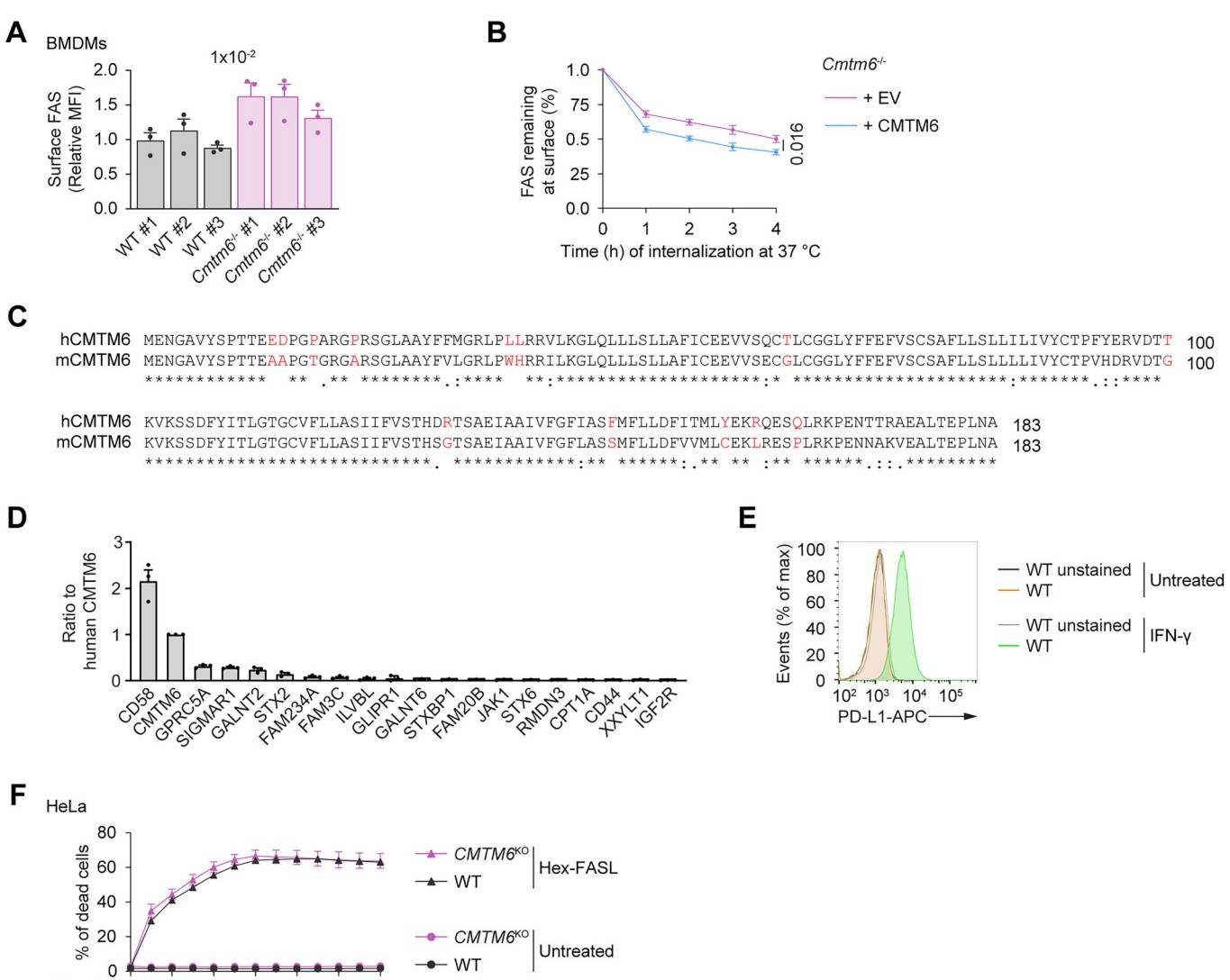

**Figure EV5. CMTM6 regulates FAS surface expression and activity in mice, but not humans.**

(A) Flow cytometry analysis of surface FAS expression in bone BMDMs isolated from 10-week-old WT and $Cmtm6^{-/-}$ mice, $n = 3$ per group. Cells from each mouse were analyzed in three separate experiments. (B) FAS internalization assay in $Cmtm6^{-/-}$ MEFs (clones #1 and #2) reconstituted with murine CMTM6 or empty vector (EV). Cells were labeled on ice with unconjugated FAS antibody and subsequently incubated for the indicated time at 37 °C. The remaining surface FAS was detected using a fluorescently labeled secondary antibody and analyzed by flow cytometry. (C) Comparison between murine and human CMTM6 sequences. (D) Mass spectrometry analysis of the SF-tagged human CMTM6 isolated from unstimulated HeLa cells via tandem affinity purification. The 20 most abundant CMTM6 interactors not detected in control EGFP-SF samples are shown, based on Top3 intensities from 3 independent experiments. (E) Flow cytometry analysis of surface PD-L1 expression in WT HeLa cells stimulated or not with human IFN-γ (100 ng/ml). (F) Cell death induction in WT or CMTM6-deficient HeLa cells stimulated or not with Hex-FASL (500 ng/ml). Cell death was monitored every 2 h using Incucyte. Data are presented as mean + SEM (A) or mean ± SEM (B, F). Data are representative of two (F), three (A, B, D), or five (E) independent experiments using two different knockout clones (B, F). One-way ANOVA (A), unpaired two-tailed t-test (B). ns not significant, MFI median fluorescent intensity. Source data are available online for this figure.

