## [Peer Review File · EMBO Reports]

CMTM6 suppresses cell-surface expression of death receptor FAS in mice but not in humans

Peter Draber, Tereza Semberova, Michaela Pribikova, Veronika Cimermanova, Tijana Trivic, Rafik Haderbache, Darina Paprckova, Luca Christen, Helena Kissiova, and Ondrej Stepanek

Corresponding author(s): Peter Draber (peter.draber@unil.ch)

Review Timeline:

Submission Date:	10th May 25
Editorial Decision:	1st Jul 25
Revision Received:	9th Nov 25
Editorial Decision:	10th Dec 25
Revision Received:	18th Dec 25
Accepted:	13th Jan 26

Transaction Report:

Dear Dr. Draber

Thank you for the submission of your research manuscript to our journal. We have now received the full set of referee reports that is copied below.

As you will see, the referees acknowledge that the findings are interesting and that the conclusions are overall supported by the data presented but they also raise a number of partially overlapping concerns and have suggestions how to further strengthen the data that need to be addressed. The concern from referee #2 regarding absence of in vivo relevance ("A key limitation is that the in vivo impact of the Cmtm6-Fas interaction is inferred...") should be addressed in the discussion, but I think that the suggestion to analyse T cell activation and survival using the available KO mice (point 2) would significantly strengthen your study and provide physiological context.

Please let me know in case you disagree, and we can discuss the exact revision requirements further, also in a video chat, if you like.

Given these constructive comments, we would like to invite you to revise your manuscript with the understanding that the referee concerns (as detailed above and in their reports) must be fully addressed and their suggestions taken on board. Please address all referee concerns in a complete point-by-point response. Acceptance of the manuscript will depend on a positive outcome of a second round of review. It is EMBO Reports policy to allow a single round of revision only and acceptance or rejection of the manuscript will therefore depend on the completeness of your responses included in the next, final version of the manuscript.

We realize that it is difficult to revise to a specific deadline. In the interest of protecting the conceptual advance provided by the work, we recommend a revision within 3 months (October 1st). Please discuss the revision progress ahead of this time with the editor if you require more time to complete the revisions.

I am also happy to discuss the revision further via e-mail or a video call, if you wish.

=====
IMPORTANT NOTE:

We perform an initial quality control of all revised manuscripts before re-review. Your manuscript will FAIL this control and the handling will be delayed IN CASE the following APPLIES:

- 1) A data availability section providing access to data deposited in public databases is missing. If you have not deposited any data, please add a sentence to the data availability section that explains that.
- 2) Your manuscript contains statistics and error bars based on $n=2$. Please use scatter blots in these cases. No statistics should be calculated if $n=2$.

=====

- 1) a .docx formatted version of the manuscript text (including legends for main figures, EV figures and tables). Please make sure that the changes are highlighted to be clearly visible.
- 2) individual production quality figure files as .eps, .tif, .jpg (one file per figure). Please download our Figure Preparation Guidelines (figure preparation pdf) from our Author Guidelines pages <https://www.embopress.org/page/journal/14693178/authorguide> for more info on how to prepare your figures.

4) a complete author checklist, which you can download from our author guidelines (<<https://www.embopress.org/page/journal/14693178/authorguide>>). Please insert information in the checklist that is also reflected in the manuscript. The completed author checklist will also be part of the RPF.

5) Please note that all corresponding authors are required to supply an ORCID ID for their name upon submission of a revised manuscript (<<https://orcid.org/>>). Please find instructions on how to link your ORCID ID to your account in our manuscript tracking system in our Author guidelines (<<https://www.embopress.org/page/journal/14693178/authorguide#authorshipguidelines>>)

6) We replaced Supplementary Information with Expanded View (EV) Figures and Tables that are collapsible/expandable online. A maximum of 5 EV Figures can be typeset. EV Figures should be cited as 'Figure EV1, Figure EV2' etc... in the text and their respective legends should be included in the main text after the legends of regular figures.

7) Before submitting your revision, primary datasets (and computer code, where appropriate) produced in this study need to be deposited in an appropriate public database (see <<https://www.embopress.org/page/journal/14693178/authorguide#dataavailability>>).

Specifically, we would kindly ask you to provide public access to the mass spectrometry datasets.

The accession numbers and database should be listed in a formal "Data Availability " section (placed after Materials & Method) that follows the model below (see also <<https://www.embopress.org/page/journal/14693178/authorguide#dataavailability>>). Please note that the Data Availability Section is restricted to new primary data that are part of this study.

Data availability

Additional information on source data and instruction on how to label the files are available <<https://www.embopress.org/page/journal/14693178/authorguide#sourcedata>>

10) Figure legends and data quantification:
The following points must be specified in each figure legend:

- the name of the statistical test used to generate error bars and P values,
- the EXACT p-values,

- the number (n) of independent experiments (please specify technical or biological replicates) underlying each data point,
- the nature of the bars and error bars (s.d., s.e.m.)

- If the data are obtained from n {less than or equal to} 5, show the individual data points in addition to the SD or SEM.
- If the data are obtained from n {less than or equal to} 2, use scatter blots showing the individual data points.

11) Our journal encourages inclusion of *data citations in the reference list* to directly cite datasets that were re-used and obtained from public databases. Data citations in the article text are distinct from normal bibliographical citations and should directly link to the database records from which the data can be accessed. In the main text, data citations are formatted as follows: "Data ref: Smith et al, 2001" or "Data ref: NCBI Sequence Read Archive PRJNA342805, 2017". In the Reference list, data citations must be labeled with "[DATASET]". A data reference must provide the database name, accession number/identifiers and a resolvable link to the landing page from which the data can be accessed at the end of the reference. Further instructions are available at <<https://www.embopress.org/page/journal/14693178/authorguide#referencesformat>>.

12) All Materials and Methods need to be described in the main text using our 'Structured Methods' format. According to this format, the Methods section includes a Reagents and Tools Table (listing key reagents, experimental models, software and relevant equipment and including their sources and relevant identifiers) followed by a Methods and Protocols section describing the methods, ideally using a step-by-step protocol format. The aim is to facilitate adoption of the methodologies across labs. Please download and fill our Reagents and Tools Table template (.docx), which you can find in our author guidelines: <https://www.embopress.org/page/journal/14693178/authorguide#structuredmethods>.

13) As part of the EMBO publication's Transparent Editorial Process, EMBO Reports publishes online a Review Process File to accompany accepted manuscripts. This File will be published in conjunction with your paper and will include the referee reports, your point-by-point response and all pertinent correspondence relating to the manuscript.

Yours sincerely,

=====

Referee #1:

CMTM6 and CMTM4, members of a four transmembrane domain containing protein family, are known to regulate the plasma membrane expression of several proteins, including PD-L1, IL17 receptor C, and EGFR. In this manuscript, the authors report

the cell death receptor FAS as a novel interactor of CMTM6 in mouse cells, but interestingly, this interaction is lost in human cells. This finding reveals species-specific features of CMTM6 that has implications on preclinical models evaluating the therapeutic potential of CMTM6. This study is well executed at the biochemical level, but functional aspects can be enhanced.

Major:

1. One major finding of the current study is that CMTM6 downregulates FAS surface expression, rendering tumor cells more resistant to FASL-induced apoptosis. This finding contrasts the reported roles of CMTM6 on stabilizing PD-L1 expression, through promoting its recycling to the plasma membrane (PMID: 28813417). Can the authors please explore the molecular mechanisms how CMTM6 regulates FAS expression/localization? For example, imaging the subcellular localization of CMTM6 and/or FAS, together with the markers for endosomes and lysosomes would provide valuable insight, as done previously for PD-L1 (PMID: 28813417).
2. This study heavily relies on co-IP experiments, with little functional data, even in the in vitro setting. Specifically, there is a lack of functional data in human cells. Can the authors please examine FASL-induced cell death differ in Hela cells with/without CMTM6 expression?
3. The manuscript would benefit from analyzing the evolutionary divergence of the FAS:CMTM6 interaction. A FAS sequence alignment across species could reveal when this interaction emerged. Given that CD58 is absent in mice, did FAS evolution track with CD58 appearance?
4. Figure 2E,F, I see actin signals in the IP samples? Does it mean that FAS binds to actin? If so how does this alter the interpretation of the data?

Minor:

1. Fig. 2A: What about other CMTM family proteins? Was CMTM6 the only member that was enriched by FAS?
2. Figure S4D presents important data that CMTM6 does not affect FAS expression in human cells. This data deserves to be shown in a main figure.

Referee #2:

In this manuscript, Semberova et al. uncover an interesting species-specific difference in the interaction between mouse *Cmtm6* and the death receptor Fas. Using quantitative proteomics and a series of well-controlled validation experiments, the authors demonstrate that mouse *Cmtm6* physically associates with Fas and suppresses its surface expression in stromal cells. Consistently, in *Cmtm6* knockout mice, Fas levels are markedly elevated at the cell surface across the major T cell subsets. Functionally, the increased surface expression of Fas in *Cmtm6*-deficient mouse MEFs leads to enhanced sensitivity to FasL-mediated apoptosis.

By contrast, the study finds that human FAS neither interacts with CMTM6 nor regulates its expression. Through elegant domain-swapping and mutagenesis experiments, the authors identify 3 amino acid differences in the extracellular/transmembrane junction that explain the loss of such interaction in human cells. This mechanistic insight is a notable strength of the study.

Overall, the findings are novel and significant for the immunology communities. CMTM6 is well known for its role in stabilizing immune checkpoint proteins PD-L1 and CD58. However, the net immunological consequences of CMTM6 loss vary depending on the model system, in part due to species-specific differences, such as the absence of a CD58 homolog in mice. The discovery that *Cmtm6* regulates Fas in mice but not in humans provides an important mechanistic insight for these divergent outcomes. It also underscores the need for caution when interpreting data from murine models, especially syngeneic tumor models, in translational contexts.

A key limitation is that the in vivo impact of the *Cmtm6*-Fas interaction is inferred rather than directly demonstrated. The authors rely primarily on cell-based assays and previously published results to provide functional context.

Overall, this manuscript is well-written, and the experimental work is carefully executed, with high-quality data presented throughout. The study provides compelling new insight into CMTM6 biology and offers a valuable caution against assuming cross-species equivalence in immune regulation. I believe that, with some minor revisions, the manuscript will be further strengthened.

Specific Comments

1. One might have expected *Cmtm6* to promote Fas surface expression as it does for PD-L1 and CD58, but here it does the opposite, which makes the finding intriguing. The authors propose a plausible explanation that *Cmtm6* may promote internalization of Fas, and its absence allows Fas to accumulate at the cell surface. The paper would be further strengthened by providing experimental support for this hypothesis. While not strictly required, the authors could, for example, assess colocalization of Fas and *Cmtm6* in endosomal/lysosomal compartments, or directly measure Fas internalization rates in wild-type vs *Cmtm6*-deficient cells.
2. Surface Fas levels are elevated across all major T cell subsets in *Cmtm6* knockout mice. This is an important finding, as it may help reconcile discrepancies in the reported functional outcomes of *Cmtm6* deficiency across different models. Given that the necessary resources (e.g., *Cmtm6* knockout mice) are available, it would be highly informative to investigate whether the increased Fas expression impacts T cell activation, survival, or function. For example, elevated Fas levels could render T cells more susceptible to activation-induced cell death, potentially impairing their functionality. Such functional assessments would provide valuable insight into the consequences of *Cmtm6* loss in the immune compartment in mouse models.

CMTM6 suppresses cell-surface expression of death receptor FAS in mice but not in humans**Semberova et al.****Responses to Reviewers*****Referee #1:***

CMTM6 and CMTM4, members of a four transmembrane domain containing protein family, are known to regulate the plasma membrane expression of several proteins, including PD-L1, IL17 receptor C, and EGFR. In this manuscript, the authors report the cell death receptor FAS as a novel interactor of CMTM6 in mouse cells, but interestingly, this interaction is lost in human cells. This finding reveals species-specific features of CMTM6 that has implications on preclinical models evaluating the therapeutic potential of CMTM6. This study is well executed at the biochemical level, but functional aspects can be enhanced.

We thank the Reviewer for assessing our work positively and for providing valuable suggestions on how to improve the manuscript further.

Major:

1. One major finding of the current study is that CMTM6 downregulates FAS surface expression, rendering tumor cells more resistant to FASL-induced apoptosis. This finding contrasts the reported roles of CMTM6 on stabilizing PD-L1 expression, through promoting its recycling to the plasma membrane (PMID: 28813417). Can the authors please explore the molecular mechanisms how CMTM6 regulates FAS expression/localization? For example, imaging the subcellular localization of CMTM6 and/or FAS, together with the markers for endosomes and lysosomes would provide valuable insight, as done previously for PD-L1 (PMID: 28813417).

The question about how CMTM6 suppresses FAS membrane expression was also raised by Reviewer #2 in question 1. Based on the suggestions from both Reviewers, we performed two types of experiments to address this issue.

First, we analyzed whether CMTM6 promotes FAS internalization. Cells were labeled on ice with anti-FAS antibody and subsequently incubated at 37°C for several hours. The remaining surface FAS was subsequently detected using a fluorescently labeled secondary antibody. *Cmtm6*^{-/-} MEFs had a substantially decreased rate of FAS internalization compared to WT cells (newly added Fig. 5G). We confirmed these data by showing that *Cmtm6*^{-/-} MEFs reconstituted with CMTM6 had enhanced FAS

internalization compared to cells transduced with EV (newly added Fig. EV5B). These data showed that CMTM6 promotes FAS internalization, which is in accord with increased surface FAS expression in CMTM6-deficient cells compared to control cells.

Subsequently, we performed an analysis of the subcellular localization of FAS and CMTM6. We expressed CMTM6-mCherry and FAS-EGFP in *Cmtm6*^{-/-} cells and showed the colocalization of both proteins in recycling endosomes, detected by staining with transferrin receptor (newly added Fig. 5H). Altogether, these data demonstrated that CMTM6 enhances FAS internalization and therefore suppresses its surface expression.

2. This study heavily relies on co-IP experiments, with little functional data, even in the in vitro setting. Specifically, there is a lack of functional data in human cells. Can the authors please examine FASL-induced cell death differ in HeLa cells with/without CMTM6 expression?

Human FAS does not interact with CMTM6 (Fig. 6A-C), and its surface expression is unchanged in HeLa WT versus HeLa *CMTM6*^{KO} (Fig. 7A). In accord, HeLa WT and *CMTM6*^{KO} cells do not differ in their sensitivity towards FASL-induced cell death (newly added Fig. EV5F). These data further strengthen our conclusion that CMTM6 suppresses FAS expression and FASL-induced cell death in mouse cells, but not in human cells.

3. The manuscript would benefit from analyzing the evolutionary divergence of the FAS:CMTM6 interaction. A FAS sequence alignment across species could reveal when this interaction emerged. Given that CD58 is absent in mice, did FAS evolution track with CD58 appearance?

To establish when the interaction between FAS and CMTM6 evolved, we first needed to find a reliable predictor of this interaction.

In our mapping of the interaction between mouse FAS and CMTM6, we showed the critical role of arginine localized just above the transmembrane domain of mouse FAS. Mutation of this residue to asparagine completely abolished the interaction of mouse FAS with CMTM6 (Fig. 6H). Interestingly, the comparison of the transmembrane and juxtamembrane regions of mouse FAS with mouse PD-L1, human PD-L1, or human CD58 proteins showed that there is very little overlap in amino acid sequence between these strong CMTM6 interactors. There was, however, one notable exception: very strong conservation of arginine above the plasma membrane (newly added Fig. 7B). To verify that this residue is universally required for CMTM6 interaction, we performed a detailed mutagenesis screen of amino acids at the boundary of the extracellular and transmembrane domains of PD-L1. This experiment showed that mutation R238A nearly abolished the interaction with CMTM6 (newly added Fig. 7C). Therefore, the emergence of arginine positioned above the plasma membrane seems as a critical requirement for CMTM6 interaction.

The analysis of the FAS transmembrane domain from various mammalian species showed that arginine above the transmembrane domain is very rare, as it is lacking even in rats. Interestingly, this

residue is not present in the Asian lineage mouse *M. caroli*, but appears in the Palearctic mouse group, which includes *M. spretus*, *M. musculus*, and *M. spicilegus* (newly added Fig 7D). These mouse species diverged approximately 3-6 million years ago, indicating that the evolution of FAS and CMTM6 interaction is a relatively recent evolutionary event.

4. Figure 2E and F, I see actin signals in the IP samples? Does it mean that FAS binds to actin? If so how does this alter the interpretation of the data?

We apologize for the confusing actin staining in immunoprecipitation (IP) samples. We employed actin staining as a standard loading control for lysates in our immunoblotting experiments. However, actin is notoriously sticky and binds directly to immunoprecipitation beads. This is well documented in the ‘Crapome database’, where it was shown using mass spectrometry that actin is a widespread contamination in immunoprecipitation experiments (Mellacheruvu et al., 2014, Nat Methods, PMID: 23921808).

In accord, we detected actin in immunoprecipitation samples when we used a control antibody, which was of the same intensity as when we used the anti-FAS antibody. This shows that detection of actin in IP samples is due to background staining, not due to specific binding to FAS. In contrast, CMTM6 was strongly isolated in samples subjected to immunoprecipitation with the anti-FAS antibody, while there was no signal using the control antibody.

We removed actin staining from panels 2E and 2F, since these experiments contain only one lysate sample separated into two IP samples using control and FAS antibody. The actin staining, therefore, did not provide any additional information.

Minor:

1. Fig. 2A: What about other CMTM family proteins? Was CMTM6 the only member that was enriched by FAS?

We observed that mouse FAS is strongly associated with CMTM6 and very weakly with CMTM4 (Fig. 1B-C). Mass spectrometry analysis of exogenously expressed mouse FAS-SF showed that CMTM6 is its strongest interactor in nonstimulated cells. We detected only a very weak signal from CMTM4, while no other CMTM family member was associated with mouse FAS-SF (Table EV2). Importantly, we could not detect an association of endogenous mouse FAS with CMTM4 in various cell lines and MEFs.

To further show that endogenous mouse FAS binds strongly to CMTM6, but not CMTM4, we isolated bone marrow-derived macrophages from WT or *Cmtm6*^{-/-} mice. We confirmed the interaction between mouse FAS and CMTM6 in WT cells (newly added Fig. 2H), and we showed that deficiency of CMTM6 promotes FAS membrane expression also in these primary cells (newly added Fig. EV5A). However, we did not detect interaction between FAS and CMTM4 in WT or *Cmtm6*^{-/-} BMDMs (newly

added Fig. 2H). Combined, these data show that in the endogenous system, FAS binds strongly to CMTM6, but not other members of the CMTM family.

2. Figure S4D presents important data that CMTM6 does not affect FAS expression in human cells. This data deserves to be shown in a main figure.

We moved this panel to the main figure (Fig. 7A).

Referee #2:

In this manuscript, Semberova et al. uncover an interesting species-specific difference in the interaction between mouse Cmtm6 and the death receptor Fas. Using quantitative proteomics and a series of well-controlled validation experiments, the authors demonstrate that mouse Cmtm6 physically associates with Fas and suppresses its surface expression in stromal cells. Consistently, in Cmtm6 knockout mice, Fas levels are markedly elevated at the cell surface across the major T cell subsets. Functionally, the increased surface expression of Fas in Cmtm6-deficient mouse MEFs leads to enhanced sensitivity to FasL-mediated apoptosis.

By contrast, the study finds that human FAS neither interacts with CMTM6 nor regulates its expression. Through elegant domain-swapping and mutagenesis experiments, the authors identify 3 amino acid differences in the extracellular/transmembrane junction that explain the loss of such interaction in human cells. This mechanistic insight is a notable strength of the study.

Overall, the findings are novel and significant for the immunology communities. CMTM6 is well known for its role in stabilizing immune checkpoint proteins PD-L1 and CD58. However, the net immunological consequences of CMTM6 loss vary depending on the model system, in part due to species-specific differences, such as the absence of a CD58 homolog in mice. The discovery that Cmtm6 regulates Fas in mice but not in humans provides an important mechanistic insight for these divergent outcomes. It also underscores the need for caution when interpreting data from murine models, especially syngeneic tumor models, in translational contexts.

A key limitation is that the in vivo impact of the Cmtm6-Fas interaction is inferred rather than directly demonstrated. The authors rely primarily on cell-based assays and previously published results to provide functional context.

Overall, this manuscript is well-written, and the experimental work is carefully executed, with high-quality data presented throughout. The study provides compelling new insight into CMTM6 biology and offers a valuable caution against assuming cross-species equivalence in immune regulation. I believe that, with some minor revisions, the manuscript will be further strengthened.

We thank the Reviewer for their positive evaluation of our data and for the suggestions on how to improve the manuscript further.

Specific Comments

1. One might have expected Cmtm6 to promote Fas surface expression as it does for PD-L1 and CD58, but here it does the opposite, which makes the finding intriguing. The authors propose a plausible explanation that Cmtm6 may promote internalization of Fas, and its absence allows Fas to accumulate at the cell surface. The paper would be further strengthened by providing experimental support for this hypothesis. While not strictly required, the authors could, for example, assess colocalization of Fas and Cmtm6 in endosomal/lysosomal compartments, or directly measure Fas internalization rates in wild-type vs Cmtm6-deficient cells.

The question about how CMTM6 suppresses FAS membrane expression was also raised by Reviewer #1 in question 1. Based on the suggestions from both Reviewers, we performed two types of experiments to address this issue.

First, we analyzed whether CMTM6 promotes FAS internalization. Cells were labeled on ice with anti-FAS antibody and subsequently incubated at 37°C for several hours. The remaining surface FAS was subsequently detected using a fluorescently labeled secondary antibody. *Cmtm6*^{-/-} MEFs had a substantially decreased rate of FAS internalization compared to WT cells (newly added Fig. 5G). We confirmed these data by showing that *Cmtm6*^{-/-} MEFs reconstituted with CMTM6 had enhanced FAS internalization compared to cells transduced with EV (newly added Fig. EV5B). These data showed that CMTM6 promotes FAS internalization, which is in accord with increased surface FAS expression in CMTM6-deficient cells compared to control cells.

Subsequently, we performed an analysis of the subcellular localization of FAS and CMTM6. We expressed CMTM6-mCherry and FAS-EGFP in *Cmtm6*^{-/-} cells and showed the colocalization of both proteins in recycling endosomes, detected by staining with transferrin receptor (newly added Fig. 5H). Altogether, these data demonstrated that CMTM6 enhances FAS internalization and therefore suppresses its surface expression.

2. Surface Fas levels are elevated across all major T cell subsets in Cmtm6 knockout mice. This is an important finding, as it may help reconcile discrepancies in the reported functional outcomes of Cmtm6 deficiency across different models. Given that the necessary resources (e.g., Cmtm6 knockout mice) are available, it would be highly informative to investigate whether the increased Fas expression impacts T cell activation, survival, or function. For example, elevated Fas levels could render T cells more susceptible to activation-induced cell death, potentially impairing their functionality. Such functional assessments would provide valuable insight into the consequences of Cmtm6 loss in the immune compartment in mouse models.

To address whether CMTM6 ablation impacts the T cell activation in vivo, we infected WT and *Cmtm6*^{-/-} mice intravenously with *Listeria monocytogenes* expressing ovalbumin (OVA) antigen (newly

added Fig. 4A). After 10 days, we isolated the spleens and analyzed T cells via flow cytometry (Fig. 4B and gating strategy in Fig. EV4A). We noted a markedly decreased proportion of effector CD4⁺ T cells (Fig. 4C), while the emergence of effector and OVA-specific CD8⁺ T cells was not significantly impacted (Fig. 4D). In contrast to CD8⁺ cells, CD4⁺ effectors are highly sensitive to FAS-mediated activation-induced cell death (AICD). Therefore, the decreased proportion of effector CD4⁺ T cells upon infection in *Cmtm6*^{-/-} mice compared to WT littermates is in accord with enhanced FAS expression and presumably increased AICD of these cells.

Importantly, we noticed that, similarly to naïve cells, effector T cells isolated from *Cmtm6*^{-/-} mice had substantially increased expression of surface FAS compared to WT mice (Fig. 4E-I). In contrast, the surface levels of FASL were not different between WT and *Cmtm6*^{-/-} mice (EV4B-F). These data further demonstrate that CMTM6 is a potent negative regulator of FAS expression in murine T cells.

Dear Dr. Draber

Thank you for the submission of your revised manuscript to EMBO reports. We have now received the full set of referee reports that is copied below.

As you will see, both referees are very positive about the study and recommend publication.

From the editorial side, there are a few things that we need before we can proceed with the official acceptance of your study:

- Please provide up to 5 keywords on the title page.
- Please rename to Conflict of Interest section to Disclosure and Competing Interests Statement.
- Regarding the Author Contributions, we now use CRediT to specify the contributions of each author in the journal submission system. Therefore, please remove the Author Contributions from the manuscript file and make sure that the author contributions in our online manuscript tracking system are correct and up-to-date. The information you specified in the system will be automatically retrieved and typeset into the article. You can enter additional information in the free text box provided, if you wish. See also our guide to authors <https://www.embopress.org/page/journal/14693178/authorguide#authorshipguidelines>.
- Figure S3 (line 809) is a wrong callout and needs correction.
- Tables EV1 - EV3 are complex datasets and should therefore be updated to Dataset EV1-Dataset EV3 in all places: source file names, legends, titles in the system, callouts in the manuscript.
- Materials and Methods should be renamed to Methods
- Please provide the specific URLs for datasets (ProteomeXchange Consortium via the PRIDE partner repository with the dataset identifiers PXD070246, PXD070298, and PXD070359) in the data availability statement.
- Please address the following points in the figure legends:
 - a) Please provide the information related to n in the legends of figures 1B.
 - b) Please define the error bars in the legends of figures 1B; 2A; 5A-G; 7A; EV-5A,B,F.
 - c) Please define the asterisk in the legend of figure 6A,B.
- Animal models: please provide the reference number of the approval of mouse work by the Czech Academy of Sciences.
- Finally, EMBO Reports papers are accompanied online by
 - A) a short (1-2 sentences) summary of the findings and their significance,
 - B) 2-3 bullet points highlighting key results and
 - C) a schematic summary figure that provides a sketch of the major findings (not a data image).Please provide the summary figure as a separate file in PNG or JPG format at a size of 550x300-600 pixels (width x height). Please note that the size is rather small and that text needs to be readable at the final size. Please send us this information along with the revised manuscript.

With kind regards,

=====

Referee #1:

The authors have addressed our concerns fully

Referee #2:

My comments have been satisfactorily addressed.

All minor editorial requests have been addressed by the authors.

Dr. Peter Draber
University of Lausanne
Department of Immunobiology
Ch. des Boveresses 155
Epalinges, Switzerland 1066
Switzerland

Dear Dr. Draber,

I am very pleased to accept your manuscript for publication in the next available issue of EMBO reports. Thank you for your contribution to our journal.

You may qualify for financial assistance for your publication charges - either via a Springer Nature fully open access agreement or an EMBO initiative. Check your eligibility: <https://link.springer.com/journal/44319/how-to-publish-with-us>

Yours sincerely,

>>> Please note that it is EMBO Reports policy for the transcript of the editorial process (containing referee reports and your response letter) to be published as an online supplement to each paper. If you do NOT want this, you will need to inform the Editorial Office via email immediately. More information is available here: <https://link.springer.com/partners/embo-press/editorial-policies#Peer%20review>